# LlaSMol: Advancing Large Language Models for Chemistry with a Large-Scale, Comprehensive, High-Quality Instruction Tuning Dataset

**Botao Yu**   **Frazier N. Baker**[*]   **Ziqi Chen**[*]   **Xia Ning**   **Huan Sun**
The Ohio State University
Columbus, OH 43210, USA
{yu.3737, baker.3239, chen.8484, ning.104, sun.397}@osu.edu

## Abstract

Chemistry plays a crucial role in many domains, such as drug discovery and material science. While large language models (LLMs) such as GPT-4 exhibit remarkable capabilities on natural language processing tasks, existing research indicates that their performance on chemistry tasks is discouragingly low. In this paper, however, we demonstrate that our developed LLMs can achieve very strong results on a comprehensive set of chemistry tasks, outperforming the most advanced GPT-4 and Claude 3 Opus by a substantial margin. To accomplish this, we propose SMolInstruct, a *large-scale*, *comprehensive*, and *high-quality* dataset for instruction tuning. It contains 14 selected chemistry tasks and over three million samples, laying a solid foundation for training and evaluating LLMs for chemistry. Using SMolInstruct, we fine-tune a set of open-source LLMs named as LlaSMol, among which, we find that Mistral serves as the best base model for chemistry tasks. Our analysis further demonstrates the critical role of the proposed dataset in driving the performance improvements.[1]

## 1 Introduction

Chemistry is a fundamental science that underpins countless aspects of modern life, ranging from drug discovery and materials science to energy production. To facilitate research and applications in this domain, deep learning models including graph neural networks (Kipf & Welling, 2017) and Transformer-based models (Vaswani et al., 2017) have been developed for various chemistry tasks such as forward reaction prediction, retrosynthesis, property prediction (Schwaller et al., 2019; Zhong et al., 2022; Chen et al., 2023; Zhou et al., 2023). However, these models are usually task-specific models, which neglect shared chemistry knowledge across tasks and can hardly be adapted to different tasks.

On the other hand, large language models (LLMs) such as GPT-4 (OpenAI, 2023), Llama series (Touvron et al., 2023a;b), and Mistral (Jiang et al., 2023) have emerged as general-purpose foundation models and demonstrate remarkable abilities on various natural language processing tasks (Chang et al., 2024; Thirunavukarasu et al., 2023; Yue et al., 2023; Zhang et al., 2023; Deng et al., 2023). However, when applied to chemistry tasks, LLMs show only limited capabilities (Jablonka et al., 2022; Guo et al., 2023; Hatakeyama-Sato et al., 2023). For example, Guo et al. (2023) conducted evaluations on eight chemistry tasks and observed that while GPT-4 outperforms other closed- and open-source LLMs, its performance is far from that of task-specific deep learning models. Particularly, they found that GPT models perform poorly when a precise understanding of SMILES (Weininger, 1988), a widely used textual representation for molecules, is required. In addition to directly applying pretrained LLMs, Fang et al. (2023) fine-tuned LLMs on an instruction tuning dataset, but their performance remains very low, far behind the state-of-the-art (SoTA) models designed and trained for specific tasks.

---

[*]Equal contribution.
[1]Our dataset and models can be found at `https://osu-nlp-group.github.io/LLM4Chem/`.

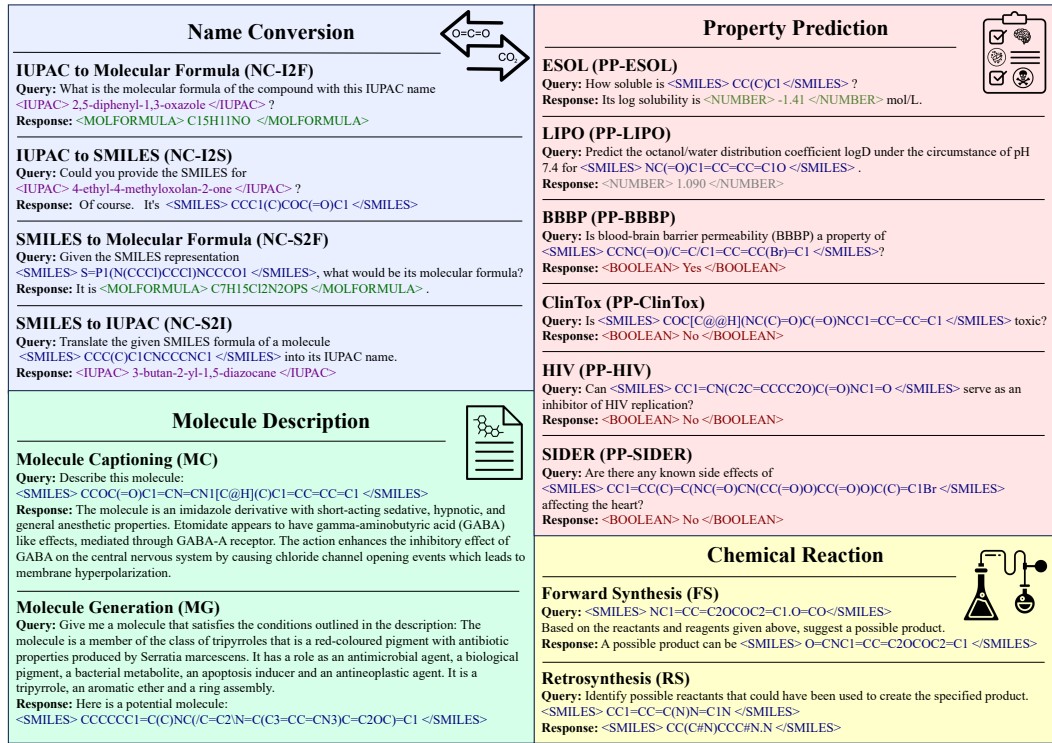

Figure 1: An overview of tasks in the proposed SMolInstruct dataset.

Given these discouraging results, some critical questions arise: *Are LLMs actually able to effectively perform chemistry tasks? Or, Are they fundamentally limited for chemistry?* In this paper, we demonstrate that our developed LLMs can achieve very strong results on a comprehensive set of chemistry tasks, substantially outperforming the most advanced GPT-4 OpenAI (2023) and Claude 3 Opus Anthropic (2024).

What makes such LLMs possible? First, we construct a large-scale, comprehensive, and high-quality dataset for instruction tuning named SMolInstruct. We incorporate tasks with meaningful applications, collect data from diverse data sources, and apply rigorous scrutiny for quality control. The resulting dataset consists of 14 tasks (illustrated in Figure 1) and over 3M samples, laying a solid foundation for training and evaluating LLMs for chemistry tasks. Based on the dataset, we build a series of LLMs for chemistry named **LlaSMol** by fine-tuning four open-source LLMs namely Galactica, Llama 2, Code Llama, and Mistral, on SMolInstruct with LoRA (Hu et al., 2022).

We conduct comprehensive experiments to evaluate our models and explore their insights, yielding some interesting findings. Firstly, among the four LlaSMol models, the Mistral-based model surpasses others by a substantial margin, showcasing the considerable influence of base models on downstream chemistry tasks. Moreover, contrast to claims made in previous work (Fang et al., 2023), using SMILES as the molecular representation achieves sufficient validity of generated molecules and better performance compared to using SELFIES (Krenn et al., 2019). Furthermore, employing canonicalized SMILES during model training and applications can alleviate learning burdens and increase performance. Finally, while instruction tuning can inject chemistry task-related knowledge into models, the dataset plays a crucial role. Our experiments demonstrate that training on our SMolInstruct leads to substantially better performance compared to training on previous dataset, emphasizing the contribution of the proposed dataset. Although LlaSMol models do not yet surpass state-of-the-art (SoTA) task-specific models that are designed and trained specifically for each individual task, they approach SoTA performance with only 0.58% of parameters being fine-tuned, suggesting their great potential for further improvements and to serve as strong foundation models for the field.

## 2 Related Work

**Task-specific Models for Chemistry.** In recent years, many deep learning models have been developed to tackle different chemistry tasks. For example, Molecular Transformer Schwaller et al. (2019) and RSMILES Zhong et al. (2022) formulate forward synthesis and retrosynthesis prediction as sequence-to-sequence translation problems. Chemformer Irwin et al. (2022) pretrains a transformer model on a large-scale SMILES dataset and fine-tunes it for various downstream tasks, such as forward synthesis and property prediction. MolT5 Edwards et al. (2022) first pretrains a T5 model on both SMILES and natural language, and then fine-tunes it to translate SMILES into natural language (i.e., molecule captioning) or vice versa (i.e., molecule generation). Graph neural networks (GNNs), which directly leverage the graph structure of the molecule Wang et al. (2023), have also shown promise in many chemistry applications, such as property prediction Yang et al. (2019); Han et al. (2023), retrosynthesis Chen et al. (2023); Somnath et al. (2021), and molecule optimization Chen et al. (2021); Zhang et al. (2022b). Recent studies Zhou et al. (2023); Zhang et al. (2022a) have shown the promise of leveraging equivariant representations of molecular 3D structures for chemistry tasks, such as property prediction Zhou et al. (2023) and docking Zhang et al. (2022a). Uni-Mol Zhou et al. (2023) incorporates this 3D information into the pretraining of a transformer model and fine-tunes it for downstream tasks. Despite their effectiveness, these models operate on single tasks and therefore cannot harness knowledge shared across diverse chemistry tasks like LLMs.

**LLMs for Chemistry.** Recent efforts have integrated LLMs with chemistry to solve key chemistry problems, which can be divided into two categories: (1) benchmark studies, and (2) fine-tuning LLMs with new datasets. Multiple benchmark studies White et al. (2023); Guo et al. (2023); Jablonka et al. (2023); Liu et al. (2023a) have evaluated the capabilities and limitations of different off-the-shelf LLMs, such as GPT-4 and Llama, on chemistry problems. For example, Guo et al. (2023) finds that these LLMs do not perform well on chemistry tasks and often produce chemically implausible outputs. These findings highlight the need for further efforts to improve LLMs via fine-tuning for chemistry tasks.

To improve LLMs for chemistry, multiple instruction tuning datasets have been developed. Mol-Instructions Fang et al. (2023) consists of 1.3M instructions for multiple small molecule tasks. However, fine-tuning on the dataset does not significantly improve LLMs' performance (Section 4.3). Drugchat Liang et al. (2023) collects an instruction tuning dataset on drug properties with 10.8K drug molecules. MolOpt-Instructions Ye et al. (2023) consists of instructions with 1M molecule pairs for molecule optimization on six properties, in which each pair has similar molecules with different properties. Recent works also develop 2D or 3D molecular graph-centric datasets and integrate the graph understanding ability into LLMs Liu et al. (2023b); Cao et al. (2023); Li et al. (2024). Compared with these datasets, SMolInstruct is much larger and covers a more diverse and comprehensive set of chemistry tasks, which enables LLMs to better understand molecule representations and learn chemistry knowledge across tasks.

## 3 SMolInstruct

This section introduces our proposed dataset SMolInstruct and its construction. Readers may refer to Appendix A for preliminaries and background.

### 3.1 Overview of SMolInstruct

SMolInstruct is a large-scale instruction tuning dataset that centers around small molecules. It contains 14 chemistry tasks, illustrated in Figure 1.

(1) We include four name conversion tasks, namely converting IUPAC name to molecular formula (NC-I2F), converting IUPAC name to SMILES (NC-I2S), converting SMILES to molecular formula (NC-S2F), and converting SMILES to IUPAC name (NC-S2I). They are designed to enable deep understanding of molecular structures and representations, which should serve as the fundamental knowledge for chemistry LLMs.

(2) Additionally, six property prediction tasks (Wu et al., 2018) are integrated, including PP-ESOL for water solubility (Mobley & Guthrie, 2014), PP-Lipo for octanol/water distribution coefficient (Poole & Poole, 2003), PP-BBBP for blood-brain barrier penetration (Martins et al., 2012), PP-ClinTox for toxicity to human body (Gayvert et al., 2016), PP-HIV for HIV replication inhibition (Institute, 2004), and PP-SIDER for side effects of drugs (Kuhn et al., 2015). These involved properties are crucial especially for drug development.

(3) Two tasks focus on the textual descriptions of molecules: molecule captioning (MC) is to generate a textual description of a given molecule, and molecule generation (MG) is to generate a molecule based on the given textual description. They require comprehensive understanding of molecules - their structures and properties, from their textual descriptions. They also bridge the gap between natural language and molecules.

(4) Lastly, two tasks revolve around chemical reaction knowledge. Forward synthesis (FS) aims to predict potential products from reactants and reagents, and retrosynthesis (RS) involves predicting potential reactants given a product. These tasks play vital roles in real-world applications (Coley et al., 2018). For example, retrosynthesis is essential for synthesis planning, while forward synthesis is used to validate retrosynthetic suggestions.

SMolInstruct contains 3.3M samples. Each sample is a query-response pair, where the query describes a task and any task-specific information (e.g., input molecule, textual description, etc.), and the response is a sentence containing the answer to the queried task. For all the tasks, unless explicitly defined in the tasks (NC-I2F, NC-I2S, NC-S2F, and NC-S2I), we use SMILES as the default representation for molecules, but also provide the SELFIES (Krenn et al., 2019) representation.

### 3.2 SMolInstruct Construction

We construct the SMolInstruct dataset by following a four-step pipeline: data collection, quality control, data splitting, and instruction construction.

**Data Collection.** After consulting domain experts and pinpointing the set of meaningful tasks (summarized in Section 3.1), we collect data for these tasks from various sources, as listed in Table 5. Specifically, for the name conversion tasks (NC-I2F, NC-I2S, NC-S2F, and NC-S2I), we leverage PubChem[2] (Kim et al., 2019), one of the most comprehensive molecule databases. Within this database, we randomly select a large set of molecule entries, and extract their IUPAC names, SMILES representations, and molecular formulas. This obtained data is then re-organized as input-output pairs for the tasks. For molecular description-related tasks (MC and MG), we utilize a combination of ChEBI-20 (Edwards et al., 2021; 2022) and Mol-Instructions (Fang et al., 2023), as they both contain high-quality molecule-text paired data. For property prediction tasks (PP-ESOL, PP-Lipo, PP-BBBP, PP-ClinTox, PP-HIV, and PP-SIDER), we employ the well-established MoleculeNet datasets (Wu et al., 2018). We select the 6 datasets from MoleculeNet that represent the essential properties for real-world applications such as drug discovery. For chemical reaction tasks (FS and RS), we collect the reaction data from USPTO-full (Lowe, 2017), which is an extensive collection encompassing over 1M reaction samples extracted from U.S. patents. All the aforementioned datasets are also widely used in previous studies (He et al., 2021; Zhong et al., 2022; Edwards et al., 2022; Irwin et al., 2022; Chen et al., 2023; Zhou et al., 2023).

**Quality Control.** To guarantee high quality, we apply rigorous scrutiny. The collected data contains many problematic and low-quality samples, which can be roughly categorized into the following three types, along with our curation methods: (1) Chemically invalid SMILES. Numerous SMILES strings are chemically invalid (e.g., deviating from the SMILES grammar, or violating chemical valence). To address this issue, we employ RDKit (RDKit, 2023), a widely used toolkit for cheminformatics, to parse molecules and detect errors. (2) Wrong or inaccurate information. Based on manual check, we observed wrong and inaccurate information recorded in the data. For instance, within the USPTO-full dataset (Lowe, 2017), we identify and correct mislabeled reactants and reagents in chemical reactions by comparing their atom mappings with products. For the MC and MG tasks, we filter out those

---

[2]https://pubchem.ncbi.nlm.nih.gov/

textual descriptions that lack pertinent, molecule-specific information, with a set of rules based on wording patterns, lengths and keywords. For PP-SIDER, we eliminate disorders with ambiguous names that could impede the creation of precise and comprehensible instructions. (3) Duplicated samples. We detect and remove them.

**Data Splitting.** Data splitting for multi-task datasets requires careful handling to prevent data leakage across tasks. For instance, FS and RS are a pair of reverse tasks, so data leakage occurs when the training set contains an FS sample for a certain chemical reaction and the test set has an RS sample for the same reaction. This can lead to biased evaluation. Therefore, we identify sample pairs across related tasks (FS and RS, MC and MG, and the four NC tasks) that correspond to the same molecules/reactions, and ensure that matched samples are placed together in either training or evaluation set. Moreover, some samples may share the same input but have different outputs. For instance, in the RS task, one product (the same input) may be synthesized from multiple sets of reactants (different outputs). If these samples are placed into both training and test set, it may lead to exaggerated performance. Therefore we ensure that samples with identical inputs are placed together either in or outside of the test set. Additionally, to achieve fair comparisons with Mol-instructions (Fang et al., 2023), for tasks shared between the two datasets (MC, MG, FS, and RS), we ensure that their training examples are not included in the test set of SMolInstruct, allowing for a direct evaluation of their models on our test set. After implementing these constraints, samples are randomly split into training/validation/test set, except for PP task samples that undergo a scaffold splitting following the canonical method (Wu et al., 2018).

**Instruction Creation.** To create query-response textual pairs for instruction tuning, we manually craft several templates, each including a query and a corresponding response, and apply GPT-4 to rephrase them. Unlike those in (Fang et al., 2023) which consist of highly formatted queries (containing three explicitly labeled parts namely instruction, input, and output) and answer-only responses (e.g., responses for FS and RS only contain answer SMILES alone, without any natural text), our templates exhibit a more natural and diverse set of formats in both queries and responses, allowing for more variations and naturalness in input-output interactions. Moreover, all the SMILES representations are canonicalized, establishing a standardized data format. In light of the dataset's inclusion of multi-type sequences (SMILES, molecular formula, numbers, etc.) beyond natural language text alone, we utilize special tags to encapsulate corresponding segments (e.g., `<SMILES>...</SMILES>` for SMILES, `<MOLFORMULA>...</MOLFORMULA>` for molecular formula, `<NUMBER>...</NUMBER>` for numbers). This design does not only explicitly inform models about the information types within the tagged content, but also facilitate answer extraction during evaluation.

For more details of dataset construction, please refer to Appendix B.2.

### 3.3 Merits of SMolInstruct

Compared to previous work (Fang et al., 2023; Liang et al., 2023; Ye et al., 2023), SMolInstruct stands out in several key aspects:

(1) **Large-Scale**. SMolInstruct consists of 3.3M samples and 1.6M distinct molecules, with a diverse range of sizes, structures, and properties (see Appendix B.1), showcasing an extensive coverage of diverse chemical knowledge.

(2) **Comprehensive**. SMolInstruct contains 4 types of chemical tasks (14 tasks in total), emerging as the most comprehensive instruction tuning dataset for small molecules. Notably, the tasks are meticulously selected to build a strong chemistry foundation model and to adapt to real-world applications.

(3) **High-Quality**. Rigorous processing steps have been implemented to exclude problematic and low-quality samples. Along with careful data splitting and canonicalization of SMILES representations, SMolInstruct stands as a high-quality resource valuable for future research.

A detailed introduction and statistics of the SMolInstruct dataset can be found in Appendix B. For a comparison with the previous work, Mol-Instructions (Fang et al., 2023), please refer to Appendix C.

# 4 Experiments

## 4.1 Our LlaSMol Models

By fine-tuning base models on the proposed SMolInstruct dataset, we create LLMs capable of performing chemistry tasks, which we name LlaSMol (**L**arge **la**nguage models on **S**mall **Mol**ecules). Specifically, we extensively consider four different LLMs as our base models, namely Galactica 6.7B (Taylor et al., 2022), Llama 2 (Touvron et al., 2023b) 7B, Code Llama (Roziere et al., 2023) 7B, and Mistral (Jiang et al., 2023) 7B, where Galactica is trained for scientific applications and has already been exposed to chemistry-related data during its pretraining, Llama 2 and Mistral are general-purpose LLMs, while Code Llama is based on Llama 2 and trained for code. We conduct instruction tuning on the proposed SMolInstruct dataset, and name the resulting models as LlaSMol$_{\text{Galactica}}$, LlaSMol$_{\text{Llama 2}}$, LlaSMol$_{\text{Code Llama}}$, and LlaSMol$_{\text{Mistral}}$, respectively. All the LlaSMol models are trained with LoRA (Hu et al., 2022), which is applied to all weight matrices in the self-attention and feedforward neural network (FFN) modules with `lora_r` and `lora_alpha` set to 16. The fine-tuning process utilizes the Huggingface Transformers library (Wolf et al., 2020). Training spans three epochs, employing the 8-bit AdamW optimizer, a learning rate of 1e-4, and a cosine scheduler. The input length for training is set to 512, which covers 99.7% of the samples. During inference, we adopt beam search as the generation strategy for simplicity.

## 4.2 Experimental Setup

**Compared Models.** We compare our LlaSMol models with two types of models:

(1) **LLMs without fine-tuning on SMolInstruct**. This type includes our four base models, namely Galactica (Taylor et al., 2022), Llama 2 (Touvron et al., 2023b), Code Llama (Roziere et al., 2023), Mistral (Jiang et al., 2023). we also benchmark against GPT-4 (OpenAI, 2023) and the more recent Claude 3 Opus (Anthropic, 2024), the current state-of-the-art (SoTA) LLMs[3]. For Llama 2, Code Llama, and Mistral, we use 1-shot, due to their poor instruction following ability; for GPT-4, we report its results under a zero-shot setting, as GPT-4 performs best on this setting in our experiments (Appendix E); for Claude 3 Opus, we report its zero-shot results as well. We also include two LLMs tuned specifically for chemistry tasks: Molinst, a Llama 2 model tuned on the Mol-Instructions dataset by Fang et al. (2023), which shares the training tasks of MC, MG, FS, and RS with LlaSMol; and ChemLLM (Zhang et al., 2024), an LLM for chemistry proposed concurrently to our work.

(2) **SoTA task-specific models.** To provide a comprehensive view of LlaSMol's performance, we present results from SoTA task-specific models. For NC-I2S and NC-S2I, we compare with STOUT (Rajan et al., 2021), an encoder-decoder model trained on SMILES-IUPAC name paired data. For NC-S2F, a task achievable with a fixed algorithm, we implement a program with RDKit (RDKit, 2023), a widely used Python toolkit for cheminformatics, and report its results. For NC-I2F where no dedicated models exist, we construct a baseline called STOUT+RDKit by aggregating STOUT for I2S conversion and RDKit for S2F conversion. For the PP tasks, our compared model is Uni-Mol (Zhou et al., 2023). It incorporates molecular 3D representations and follows a pretraining and fine-tuning paradigm. Following its original settings, we fine-tune the model on our SMolInstruct dataset with its pretrained checkpoint. In the case of MC and MG, we compare with MolT5 (Edwards et al., 2022) and directly use their released checkpoint. The reasons why we do not use our re-trained model are: (1) we were unable to reproduce results close to those reported in the paper as no original code was provided; and (2) we take great care to ensure that our test set is devoid of training examples used by MolT5, ensuring fairness in the evaluation. Lastly, regarding FS and RS, we re-train RSMILES (Zhong et al., 2022) and Molecular Transformer (Schwaller et al., 2019) for the two tasks, respectively, following their reported settings. Both of the models are transformer encoder-decoder models (Vaswani et al., 2017), specifically adapted for the FS and RS tasks.

---

[3] Due to resource limitations, we evaluate GPT-4 and Claude 3 Opus on at most 500 test samples for each task.

Table 1: Results on name conversion (NC) and property prediction (PP) tasks. Metrics EM, Valid, and Acc are in percentage.

| Model | NC | | | | | PP | | | | | |
| | I2F | I2S | | S2F | S2I | ESOL | Lipo | BBBP | Clintox | HIV | SIDER |
| | EM | EM | Valid | EM | EM | RMSE↓ | RMSE↓ | Acc | Acc | Acc | Acc |
|---|---|---|---|---|---|---|---|---|---|---|---|
| **Task-Specific, Non-LLM Based Models** | | | | | | | | | | | |
| SoTA | 97.9 | 73.5 | 99.4 | 100.0 | 56.5 | 0.819 | 0.612 | 85.3 | 92.4 | 97.0 | 70.0 |
| **Existing LLMs without fine-tuning on SMolInstruct** | | | | | | | | | | | |
| GPT-4 | 8.7 | 3.3 | 84.2 | 4.8 | 0.0 | 2.570 | 1.545 | 62.9 | 50.0 | 59.6 | 57.6 |
| Claude 3 Opus | 34.6 | 17.7 | 90.2 | 9.2 | 0.0 | **1.036** | 1.194 | **75.1** | 41.7 | 76.4 | 67.0 |
| Galactica | 9.1 | 9.7 | 95.6 | 0.0 | 0.0 | 4.184 | 2.979 | 69.0 | 92.4 | **96.7** | 68.1 |
| Llama 2 | 0.0 | 0.0 | 18.3 | 0.0 | 0.0 | 3.287 | 1.634 | 58.9 | 45.1 | 93.3 | 61.9 |
| Code Llama | 0.0 | 0.0 | 81.0 | 0.0 | 0.0 | 3.483 | 1.733 | 58.9 | 85.4 | 91.8 | 60.2 |
| Mistral | 0.0 | 0.0 | 40.3 | 0.0 | 0.0 | 3.079 | 1.730 | 40.6 | 15.3 | 7.1 | 38.1 |
| Molinst (chemistry LLM) | 0.0 | 0.0 | 96.2 | 0.0 | 0.0 | 2.271 | 1.691 | 60.9 | 6.3 | 4.5 | 52.4 |
| ChemLLM (chemistry LLM) | 0.8 | 0.3 | 3.9 | 0.0 | 0.0 | 1.946 | 1.797 | 22.3 | 75.7 | 72.9 | 32.6 |
| **Our LlaSMol Series** | | | | | | | | | | | |
| LlaSMol$_{\text{Galactica}}$ | 83.2 | 58.7 | 99.4 | 91.2 | 18.3 | 1.959 | 1.213 | 69.0 | **93.1** | **96.7** | 70.1 |
| LlaSMol$_{\text{Llama 2}}$ | 73.8 | 46.6 | 99.0 | 87.0 | 12.9 | 2.791 | 1.338 | 69.0 | 92.4 | **96.7** | 68.7 |
| LlaSMol$_{\text{Code Llama}}$ | 75.4 | 49.9 | 99.3 | 88.6 | 15.5 | 2.959 | 1.203 | 69.0 | **93.1** | **96.7** | 69.9 |
| LlaSMol$_{\text{Mistral}}$ | **87.9** | **70.1** | **99.6** | **93.2** | **29.0** | 1.150 | **1.010** | 74.6 | **93.1** | **96.7** | **70.7** |

**Evaluation Metrics.** We employ metrics commonly used in previous work (Schwaller et al., 2019; Zhong et al., 2022; Fang et al., 2023; Zhou et al., 2023; Chen et al., 2023), which include: (1) **Exact Match (EM)**, indicating the proportion of predicted results that exactly match the gold standards. (2) **Fingerprint Tanimoto Similarity (FTS)**, quantifying structural similarities between molecules using Tanimoto similarities of their Morgan fingerprints (Morgan, 1965). (3) **METEOR score**, a comprehensive text-based metric considering both exact matches and semantic similarity (Lavie & Agarwal, 2007) for the MC task. (4) **Root Mean Square Error (RMSE)**, measuring the square root of the average squared differences between predicted and actual values for the PP-ESOL and PP-Lipo tasks (5) **Accuracy (Acc)**, the ratio of correct predictions for the binary classification tasks (PP-BBBP, PP-ClinTox, PP-HIV, and PP-SIDER). (6) **Validity (Valid)**, the ratio of valid predictions following SMILES grammar and chemical valence rules for tasks with SMILES outputs (NC-I2S, MG, FS, and RS). For all the metrics except RMSE, higher values indicate better performance.

### 4.3 Main Results

Table 1 and 2 show the performance of different models on SMolInstruct. We make the following key observations:

(1) **Among all the LLMs, our LlaSMol models demonstrate the best performance, underscoring the effectiveness of the proposed SMolInstruct dataset and fine-tuning**. Specifically, compared to the base models (Galactica, Llama 2, Code Llama, and Mistral), LlaSMol models exhibit substantial performance improvements, which highlights the effectiveness of SMolInstruct in enhancing the understanding of molecular representations and the task-related knowledge, and signifies the effective learning of chemistry-related tasks by LLMs. Furthermore, LlaSMol substantially outperforms GPT-4 on all the tasks and Claude 3 Opus on most tasks, despite their larger parameter size. LlaSMol also surpasses the two chemistry LLMs namely ChemLLM[4], which is similarly trained on chemistry instruction data. and Molinst. Notably, LlaSMol$_{\text{Llama 2}}$, which uses the same base model and LoRA setting as Molinst, outperforms it even on the shared training tasks (MC, MG, FS, and RS). This finding highlights the benefits of our dataset.

(2) **Our four LlaSMol models show substantial differences in their performance, emphasizing the considerable impact of base models on downstream tasks**. Despite sharing identical training, inference settings, and comparable model sizes, LlaSMol$_{\text{Mistral}}$ consistently outperforms LlaSMol$_{\text{Llama 2}}$ by a substantial margin, highlighting Mistral's potential on chemistry tasks. In addition, LlaSMol$_{\text{Code Llama}}$ exhibits better performance than LlaSMol$_{\text{Llama 2}}$

---

[4]Since its dataset and evaluation details are not available, we cannot provide more analysis.

Table 2: Results on molecule captioning (MC), molecule generation (MG), forward synthesis (FS), and retrosynthesis (RS). Metrics EM, FTS, and Valid are in percentage.

| Model | MC | MG | | | FS | | | RS | | |
|---|---|---|---|---|---|---|---|---|---|---|
| | METEOR | EM | FTS | Valid | EM | FTS | Valid | EM | FTS | Valid |
| **Task-Specific, Non-LLM Based Models** | | | | | | | | | | |
| SoTA | 0.515 | 31.7 | 73.2 | 95.3 | 78.7 | 92.2 | 100.0 | 47.0 | 77.5 | 99.7 |
| **Existing LLMs Without Fine-Tuning on SMolInstruct** | | | | | | | | | | |
| GPT-4 | 0.188 | 6.4 | 42.6 | 81.4 | 1.6 | 40.5 | 87.0 | 0.0 | 33.4 | 42.6 |
| Claude 3 Opus | 0.219 | 12.3 | 57.6 | 92.6 | 3.7 | 45.7 | 97.0 | 1.1 | 46.2 | 94.8 |
| Galactica | 0.050 | 0.0 | 11.6 | 94.7 | 0.0 | 25.9 | 83.7 | 0.0 | 34.6 | 93.0 |
| Llama 2 | 0.150 | 0.0 | 4.8 | 93.5 | 0.0 | 13.7 | 97.7 | 0.0 | 27.5 | 87.7 |
| Code Llama | 0.143 | 0.0 | 8.5 | 95.2 | 0.0 | 15.8 | 99.6 | 0.0 | 25.3 | 97.1 |
| Mistral | 0.193 | 0.0 | 9.0 | 35.9 | 0.0 | 19.9 | 95.8 | 0.0 | 24.2 | 98.0 |
| Molinst (chemistry LLM) | 0.124 | 6.0 | 43.6 | 84.8 | 2.1 | 31.7 | 99.8 | 5.7 | 48.0 | 97.8 |
| ChemLLM (chemistry LLM) | 0.050 | 0.9 | 14.3 | 4.3 | 0.0 | 1.6 | 38.5 | 0.0 | 2.9 | 10.9 |
| **Our LlaSMol Series** | | | | | | | | | | |
| LlaSMol$_{Galactica}$ | 0.394 | 7.7 | 52.2 | 99.6 | 53.1 | 79.9 | 99.7 | 25.7 | 67.0 | 99.9 |
| LlaSMol$_{Llama 2}$ | 0.377 | 6.4 | 47.1 | 99.6 | 47.1 | 76.9 | **99.8** | 22.5 | 65.2 | 99.9 |
| LlaSMol$_{Code Llama}$ | 0.366 | 6.5 | 46.6 | **99.7** | 52.0 | 79.2 | **99.8** | 25.7 | 66.7 | **100.0** |
| LlaSMol$_{Mistral}$ | **0.452** | **19.2** | **61.7** | 99.7 | **63.3** | **84.9** | **99.8** | **32.9** | **70.4** | **100.0** |

Table 3: Ablation study results on NC and PP. Metrics EM, Valid, and Acc are in percentage. Orange indicates better results than LlaSMol$_{Mistral}$; blue indicates worse.

| Model | NC | | | | | PP | | | | | |
|---|---|---|---|---|---|---|---|---|---|---|---|
| | I2F | I2S | | S2F | S2I | ESOL | Lipo | BBBP | Clintox | HIV | SIDER |
| | EM | EM | Valid | EM | EM | RMSE↓ | RMSE↓ | Acc | Acc | Acc | Acc |
| LlaSMol$_{Mistral}$ | 87.9 | 70.1 | 99.6 | 93.2 | 29.0 | 1.150 | 1.010 | 74.6 | 93.1 | 96.7 | 70.7 |
| w/o canonical | 88.5 | 67.2 | 99.6 | 93.4 | 24.5 | 1.224 | 1.072 | 71.6 | 93.1 | 96.8 | 70.3 |
| using SELFIES | 86.9 | 47.7 | 100.0 | 94.7 | 19.7 | 1.456 | 1.106 | 69.5 | 91.7 | 96.5 | 64.4 |
| train on Mol-Instructions | 0.0 | 0.0 | 75.2 | 0.0 | 0.0 | 4.416 | 2.282 | 0.0 | 0.0 | 2.6 | 0.4 |

on most tasks, indicating a potential synergy between programming language knowledge in Code Llama and molecular representations. Furthermore, LlaSMol$_{Galactica}$ outperforms LlaSMol$_{Llama 2}$, and LlaSMol$_{Code Llama}$ in most cases, suggesting the benefits of pretraining on chemistry-related documents.

(3) **Although LlaSMol models do not outperform SoTA models, they demonstrate considerable potential for further improvements**. Specifically, LlaSMol$_{Mistral}$ surpasses the SoTA models on PP-Clintox and PP-SIDER, but has yet to achieve the success on other tasks. However, LlaSMol has greatly narrowed the performance gap between LLMs and SoTA task-specific models, compared to previous efforts (Fang et al., 2023; Zhang et al., 2024). Remarkably, LlaSMol$_{Mistral}$ attains such performance with only a small proportion of its parameters fine-tuned (approximately 41.9M, 0.58% of its parameters). As shown in Appendix F.2, increasing the number of trainable parameters can substantially boost performance, suggesting that LlaSMol$_{Mistral}$ has immense potential to surpass task-specific models through more extensive fine-tuning and serve as a strong foundation model for chemistry applications.

## 4.4 Ablation Study

To investigate the advantages of SMolInstruct, we conduct an ablation study by comparing LlaSMol$_{Mistral}$ with the following variants: (1) **w/o canonical**, which uses uncanonicalized SMILES, to examine the benefits of canonicalization. (2) **using SELFIES**, which uses SELFIES Krenn et al. (2019) instead of SMILES to explore their differences. (3) **train on Mol-Instructions**, which is trained on Mol-Instructions (Fang et al., 2023), to compare the performance improvements of our dataset against the previously proposed dataset.

The results in Table 3 and Table 4 lead to the following observations: (1) The "w/o canonical" model underperforms LlaSMol$_{Mistral}$ on most tasks, with a substantial performance drop on FS and RS. This suggests that canonicalizing SMILES can reduce learning difficulty and improve performance. As canonicalization can be easily performed using fixed algorithms

Table 4: Ablation study results on MC, MG, FS, and RS. Metrics EM, FTS, and Valid are in percentage. Orange indicates better results than LlaSMol$_{Mistral}$; blue indicates worse.

| Model | MC | MG | | | FS | | | RS | | |
|---|---|---|---|---|---|---|---|---|---|---|
| | METEOR | EM | FTS | Valid | EM | FTS | Valid | EM | FTS | Valid |
| LlaSMol$_{Mistral}$ | 0.452 | 19.2 | 61.7 | 99.7 | 63.3 | 84.9 | 99.8 | 32.9 | 70.4 | 100.0 |
| w/o canonical | 0.457 | 16.8 | 60.2 | 99.1 | 53.7 | 80.8 | 99.9 | 23.8 | 67.4 | 99.9 |
| using SELFIES | 0.466 | 16.2 | 58.6 | 99.9 | 40.4 | 74.0 | 100.0 | 25.6 | 66.0 | 99.9 |
| train on Mol-Instructions | 0.195 | 6.1 | 46.1 | 88.2 | 3.9 | 37.1 | 78.3 | 7.4 | 52.6 | 76.7 |

before feeding into models, we recommend using canonical SMILES when training and applying LLMs for chemistry. (2) While using SELFIES slightly improves the validity of generated molecules, which aligns with the motivation behind SELFIES (Krenn et al., 2019), the validity of using SMILES is also sufficiently high. Moreover, using SELFIES results in worse performance on most tasks, possibly due to SELFIES being typically longer than SMILES, making it more difficult for the model to accurately understand and generate. Therefore, using SELFIES over SMILES may not be necessary, contrast to claims made in previous work (Krenn et al., 2019; Fang et al., 2023). (3) Despite using identical base models and training settings, the model trained on Mol-Instructions (Fang et al., 2023) performs much worse than LlaSMol$_{Mistral}$ trained on SMolInstruct even on the shared tasks (MC, MG, FS, and RS). This demonstrates the superiority of our dataset. A detailed comparison with Mol-Instructions can be found in Appendix C.

To gain deeper insights into the models' performance and behavior, we conduct further analytical experiments: (1) To investigate the synergistic effects among different tasks, we evaluate models trained on a single task and models with certain tasks removed. The results demonstrate multiple-task training outperforms single-task training, indicating its benefits. However, each task generally does not heavily rely on the presence of other tasks, suggesting a degree of independence among them. (2) To investigate the influence of LoRA (Hu et al., 2022) settings, we vary the involved LoRA modules. We observe that adding LoRA modules (and trainable parameters) leads to a substantial boost in performance, indicates the models' great potential for further improvements if with larger-scale fine-tuning. Please refer to Appendix F for more details.

## 5   Conclusion

While LLMs show promise as versatile assistants, their performance on chemistry-related tasks remains notably subpar. To address this issue, we introduces SMolInstruct, a large-scale, comprehensive, and high-quality instruction tuning dataset. It comprises 14 tasks highly relevant to real-world applications and contains over 3M rigorously curated samples. Using SMolInstruct, we develop LlaSMol, a series of LLMs for performing chemistry tasks. Our experiments demonstrate LlaSMol's superiority over existing LLMs, and highlight SMolInstruct's crucial role in boosting the performance. Further analytical experiments also provide significant insights towards developing LLMs for chemistry and science.

However, this work has the following limitations. First, the evaluations for the MC and MG tasks cannot accurately assess models' abilities to generate chemically correct descriptions and molecules. Since the definition of molecular descriptions remain ambiguous and the available data is limited, it is challenging to assess whether the generated descriptions or molecules are accurate and correct. Second, this work does not delve into the models' generalization capabilities beyond the trained tasks. While we recognize the importance of such capabilities, how to meaningfully test generalization abilities is nontrivial and needs careful design, which falls outside the purview of this work. Third, our models do not yet outperform SoTA task-specific models, possibly due to the small ratio of trainable parameters or suboptimal training procedures. Nevertheless, we propose a high-quality instruction tuning dataset, demonstrate its effectiveness, and gain deeper insights, which we hope can be valuable for future research. We will try to address the aforementioned limitations in our future work.

## Ethics Statement

Despite our best efforts to maintain the high quality of the SMolInstruct dataset and the integrity of the LlaSMol models, we cannot guarantee that the dataset is free of inaccurate, incorrect, or harmful content, nor can we prevent the models from generating such content. Users should engage with our dataset and models at their own discretion and uphold the highest ethical standards in their use.

## Acknowledgement

The authors would thank colleagues from the OSU NLP group and the OSU Ning Lab for constructive feedback. This research was supported in part by NSF IIS-2133650, NIH 1R01LM014385-01, and NSF CAREER #1942980, as well as Ohio Supercomputer Center (Ohio Supercomputer Center, 1987). The views and conclusions contained herein are those of the authors and should not be interpreted as representing the official policies, either expressed or implied, of the U.S. government. The U.S. Government is authorized to reproduce and distribute reprints for Government purposes notwithstanding any copyright notice herein.

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

# Table of Contents in Appendix

## A    Preliminaries

Molecules form the basis of chemistry, which fundamentally determines the properties and behaviors of most substances. A molecule is a group of atoms held together by chemical bonds (Brown, 2018). In this paper, we focus on small molecules, which typically have no more than 100 atoms and a low molecular weight under 1,500 Daltons (Lenci & Trabocchi, 2020). Small molecules perform many important functions, such as signaling in cellular biology (McNerney & Styczynski, 2018), pest control in agriculture (Burns et al., 2006), micronutrients in nutrition (Chen et al., 2022), and drug therapy in medicine (Lenci & Trabocchi, 2020). Given the importance of small molecules, it is essential to integrate LLMs into the study of small molecules to further advance their design or development.

Molecules can be represented in multiple ways, such as SMILES strings, IUPAC names, and molecular formulas. SMILES strings use a sequence of symbols to encode the 2D structures of molecules (Weininger, 1988). A molecule can have multiple SMILES strings; a *canonical* SMILES for the molecule is unique and deterministic. For example, the canonical SMILES representation of glucose is "C(C1C(C(C(C(O1)O)O)O)O)O". SELFIES (Krenn et al., 2019) is an alternative representation to SMILES that also uses a sequence of symbols to denote molecular structures. Its key advantage is robustness, as every SELFIES string is guaranteed to correspond to a valid molecule. The SELFIES representation corresponding to the above SMILES representation of glucose is "[C][Branch2][Ring1][Branch1][C][C][Branch1][S][C][Branch1][N][C][Branch1][Branch2] [C][Branch1][Ring2][O][Ring1][=Branch1][O][O][O][O][O]". Molecular formulas represent a molecule by enumerating the type and number of atoms in the molecule (Solomons et al., 2022). For example, the molecular formula for glucose is "$C_6H_{12}O_6$". IUPAC names are formal names based on natural language elements, which follow the systematic rules set by the International Union of Preferred and Applied Chemistry (IUPAC) (Favre & Powell, 2014). These names are derived from the structures and functional groups of molecules, and are intended to be human-readable. For example, the IUPAC name for glucose is "(3R,4S,5S,6R)-6-(hydroxymethyl)oxane-2,3,4,5-tetrol".

Molecules are one of the fundamental units of chemistry that participate in reactions (Brown, 2018). A *reaction* is a process which converts input molecules (*reactants*) into output molecules (*products*) through the breaking and forming of chemical bonds. Other molecules (*reagents*) may be present to enhance or facilitate the reaction.

## B    Details of SMolInstruct

In this section, we introduce the details of our proposed dataset SMolInstruct, including statistics and construction details.

### B.1    The statistics of SMolInstruct

Table 5 shows the statistics of SMolInstruct. It contains 4 types of altogether 14 tasks, which are selected to be meaningful and useful. There are about 3.3M samples, and each of them is a distinct sample. In other words, there does not exist a pair of samples who share the same chemical information (i.e., the core input and output information, such as input molecules and output molecules), but with the same or different natural language templates (i.e., the task description in the query and the sentence templates in the response). When needed, one can easily create more instruction tuning samples by combining one piece of chemical information with multiple natural language templates. All in all, SMolInstruct can serve as a good benchmark for training and evaluating LLMs on various chemistry tasks.

To know more about the diversity of SMolInstruct, we conduct a statistics on the molecules. Altogether, there exist 1.6M distinct molecules, and several important statistical values are shown in Figure 2. Specifically, **Bertz complexity** is a topological index that measures the complexity of molecules based on the number and types of bonds and atoms. **Atom count** shows the number of atoms in a molecule, and it represents the size of a molecule. **Molecular weight** is the sum of the atomic weights of the atoms in a molecule. And **ring count** shows

Table 5: The statistics of SMolInstruct. "Qry." and "Resp." are average lengths of queries and responses, respectively.

| Task | Task abbr. | #Train | #Valid | #Test | #All | Qry. | Resp. |
|---|---|---|---|---|---|---|---|
| **Name Conversion**. Data Source: PubChem | | | | | | | |
| IUPAC to Molecular Formula | NC-I2F | 300,000 | 1,497 | 2,993 | 304,490 | 84 | 25 |
| IUPAC to SMILES | NC-I2S | 299,890 | 1,496 | 2,993 | 304,379 | 82 | 59 |
| SMILES to Molecular Formula | NC-S2F | 299,890 | 1,496 | 2,993 | 304,379 | 68 | 26 |
| SMILES to IUPAC | NC-S2I | 299,890 | 1,496 | 2,993 | 304,379 | 72 | 68 |
| **Property Prediction**. Data Source: MoleculeNet | | | | | | | |
| ESOL | PP-ESOL | 888 | 111 | 112 | 1,111 | 43 | 22 |
| Lipo | PP-Lipo | 3,360 | 420 | 420 | 4,200 | 80 | 11 |
| BBBP | PP-BBBP | 1,569 | 196 | 197 | 1,962 | 68 | 11 |
| ClinTox | PP-ClinTox | 1,144 | 143 | 144 | 1,431 | 69 | 11 |
| HIV | PP-HIV | 32,864 | 4,104 | 4,107 | 41,075 | 63 | 11 |
| SIDER | PP-SIDER | 22,820 | 2,860 | 2,860 | 28,540 | 82 | 11 |
| **Molecule Description**. Data Source: Mol-Instructions, ChEBI-20 | | | | | | | |
| Molecule Captioning | MC | 56,498 | 1,269 | 2,538 | 60,305 | 83 | 102 |
| Molecule Generation | MG | 56,498 | 1,269 | 2,493 | 60,260 | 117 | 75 |
| **Chemical Reaction**. Data Source: USPTO-full | | | | | | | |
| Forward Synthesis | FS | 971,809 | 2,049 | 4,062 | 977,920 | 98 | 52 |
| Retrosynthesis | RS | 941,735 | 2,092 | 4,156 | 947,983 | 77 | 70 |
| **Overall** | | 3,288,855 | 20,498 | 33,061 | 3,342,414 | 83 | 55 |

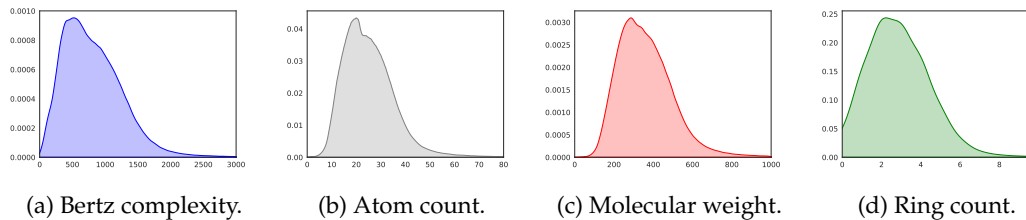

(a) Bertz complexity.     (b) Atom count.     (c) Molecular weight.     (d) Ring count.

Figure 2: The statistics of molecules in SMolInstruct, with the long tail parts removed for a clear presentation.

the number of rings in the molecular structures. As we can see, the values varies much, showing a extensive coverage in terms of complexity, size, and structure. Notably, when compared to Mol-Instructions (Fang et al., 2023), molecules in SMolInstruct show a higher complexity and diversity, which indicates that SMolInstruct is more comprehensive and complicated than Mol-Instructions. The scale, complexity, and diversity of SMolInstruct makes it well-suited for learning chemistry LLMs.

## B.2 Details of Dataset Construction

Dataset construction involves four key steps (Section 3.2): data collection, quality control, data splitting, and instruction creation. This section provides task-specific details, omitting the common steps of canonicalizing SMILES/SELFIES and verbalizing information into query and response sentences, which have been introduced in Section 3.2.

**Name Conversion (NC).** The raw data for name conversion is collected from PubChem (Kim et al., 2019). Approximately 300k molecule/compound entries are randomly selected from the database, and their SMILES, IUPAC names, and molecular formulas are extracted. Entries with incomplete or missing information in these three domains are discarded. Finally, the SMILES, IUPAC names, molecular formulas are paired to create samples for the four name conversion tasks.

**Property Prediction (PP).** The raw data for property prediction is sourced from MoleculeNet (Wu et al., 2018). Out of its 16 core datasets[5], we select 6 that are only related to small molecules and are useful especially in drug discovery. Answers for regression tasks (e.g., ESOL and Lipo) are formulated as strings of numbers, and answers for binary classification tasks (e.g., BBBP and SIDER) are formulated as "Yes" or "No".

**Molecule Captioning (MC) and Molecule Generation (MG).** The raw data is collected from ChEBI-20 (Edwards et al., 2021; 2022) and Mol-Instructions (Fang et al., 2023). Despite the large number of samples in Mol-Instructions, many are found to be of low quality. For example, numerous molecular descriptions end with the ambiguous phrase "with data available", while others are overly general, making it difficult to generate a specific molecule based on the description. To ensure data quality, regular expressions and heuristic rules are employed to filter out low-quality samples.

**Forward Synthesis (FS).** USPTO-full (Lowe, 2017), one of the most comprehensive chemical reaction datasets, serve as the data source. The following processing steps are performed to clean the data: (1) Reactants and reagents are combined as input, and the product(s) serve as output, consistent with other datasets such as Mol-Instructions (Fang et al., 2023). (2) Duplicate chemicals in both input and output are removed to avoid redundancy. (3) If a chemical appears in both input and output, it is removed from the output to maintain data integrity. (4) Products in the output containing fewer than 5 molecules are considered non-main products and excluded. (5) If the above steps result in an empty output, the entire sample is discarded.

**Retrosynthesis (RS).** The data is also sourced from USPTO-full (Lowe, 2017), with the product as input and the reactants (excluding reagents) as output. During data exploration, we observe instances where reactants are mislabeled as reagents and vice versa. To address this issue, we compare the atom mapping numbers of the reactants and reagents with the products and relabel them accordingly. Subsequently, we apply the following processing steps: (1) Duplicate chemicals in both input and output are removed. (2) If a chemical appears in both input and output, it is removed from the input. (3) Products in the input containing fewer than 5 molecules are excluded. (4) In cases where multiple products exist in the input, the reaction is split into multiple samples, with each product serving as the input once. (5) If the above steps result in an empty input, the entire sample is discarded.

For all the tasks, samples containing invalid SMILES strings (i.e., those that cannot be parsed into a valid molecule with RDKit(RDKit, 2023)) are discarded, and duplicate samples are removed to avoid redundancy. Finally, since some molecules contain multiple components and they are separated by dots in SMILES, which is the same delimiter used to separate different reactants/reagents/products in FS and RS, the dots in SMILES strings for NC, PP, MC, and MG are replaced with semicolons to differentiate between these two usages.

## C  Comparison with Mol-Instructions

In this section, we present a comprehensive comparison between our work and Mol-Instructions (Fang et al., 2023).

We begin by comparing our dataset, SMolInstruct, with the Mol-Instructions dataset. While Mol-Instructions covers a broader scope (including molecule-oriented, protein-oriented, and biomolecular text instructions), SMolInstruct focuses exclusively on small molecules, providing a deeper and more comprehensive exploration of this domain.

If focusing on the molecule-related data, as shown in Table 6, SMolInstruct is a larger, more comprehensive, and higher-quality dataset. It incorporates more tasks, samples, and molecular representations, and involves more careful curation. Both datasets share the tasks of MC, MG, FS, and RS. Although SMolInstruct has fewer samples for MC and MG, the included samples are of higher quality (see Appendix B.2). Furthermore, SMolInstruct contains substantially more samples for FS and RS, which have been carefully cleaned and processed. Additionally, SMolInstruct incorporates four NC tasks to facilitate the

---

[5] https://moleculenet.org/datasets-1

understanding of various molecular representations. Unlike Mol-Instructions, we do not include the reagent prediction task mainly due to the lack of sufficient high-quality data (Andronov et al., 2023) and the limited practicality of this task in real world applications.

Beyond the dataset, our work makes contributions to the exploration of chemistry LLMs. While Fang et al. (2023) primarily focus on the dataset itself and provide a preliminary exploration of the models, we conduct comprehensive experiments to investigate the abilities of LLMs in the chemistry domain. Our experiments in Section 4 demonstrates that our LlaSMol models achieves superior performance compared to the LLMs trained on Mol-Instructions and the strongest LLMs such as GPT-4 and Claude 3 Opus, greatly diminishing the gap between LLMs and SoTA task-specific models. Moreover, we provides valuable insights about multi-task training, LoRA (Hu et al., 2022) settings, and other aspects that could be helpful for future research in this field.

Table 6: Comparison between Mol-Instructions (the molecule-oriented part) (Fang et al., 2023) and our SMolInstruct.

|  |  | Mol-Instructions | SMolInstruct (ours) |
|---|---|---|---|
| Tasks | name conversion | ✗No this task. | ✓1.2M samples |
|  | property prediction | ✓362.1k samples on HOMO/LUMO energy. | ✓78.3k samples on 6 useful properties. |
|  | molecule captioning | ✓298.3k samples. | ✓60.3k samples. |
|  | molecule generation | ✓298.3k samples. | ✓60.3k samples. |
|  | forward synthesis | ✓125.4k samples. | ✓977.9k samples. |
|  | retrosynthesis | ✓129.7k samples. | ✓948.0k samples. |
|  | reagent prediction | ✓125.4k samples. | ✗Not included due to its insufficient data and limited practicality. |
| #Distinct samples |  | 1.2M | 3.3M |
| #Samples |  | 1.3M | 3.3M |
| Molecular representations |  | SELFIES. | Supports SMILES (default) and SELFIES, also involves IUPAC names and molecular formula in the NC tasks. |
| Data splitting |  | Provides test set, while train/validation sets are not explicitly split. | Carefully split into train/validation/test set, removing potential data leakage (see Section 3.2) |
| Canonicalization |  | No. | Yes, all the SMILES/SELFIES representations are canonicalized, providing a standardized data format. |
| Complexity and diversity of molecules |  | lower. | Higher (see Appendix B.1). |

## D  Details of Experimental Setup

In this section, we introduce the details of our experimental setups, including the training and inference details of our LlaSMol models and the compared models. We also give detailed explanations of the metrics used in Section 4.3, as well extra metrics that we will use in Appendix E.

### D.1 LlaSMol Models

The base models used for developing LlaSMol are Galactica[6] (Taylor et al., 2022), Llama 2[7] (Touvron et al., 2023b), Code Llama[8] (Roziere et al., 2023) and Mistral[9] (Jiang et al., 2023). We conduct instruction tuning on our SMolInstruct, and the resulting models are called named as $LlaSMol_{Galactica}$, $LlaSMol_{Llama\ 2}$, $LlaSMol_{Code\ Llama}$, and $LlaSMol_{Mistral}$, respectively. Expect for being based on different base models, their training and evaluation configurations are identical, as described as follows.

We used LoRA (Hu et al., 2022) during training, which is applied to all linear layers in the self-attention and FFN modules with `lora_r` and `lora_alpha` set to 16. With the 8-bit AdamW optimizer, a learning rate of 1e-4, and a cosine scheduler, we train each model for three epochs. The input length is set to 512, and sequences longer than 512 are truncated.

During inference, we adopt beam search as the generation strategy for simplicity. Due to the need of evaluations on the top-$k$ predicted answers (as in Appendix E, where $k$ varies for different tasks, we generate different numbers of sequences for different tasks by setting the `num_return_sequences` argument in the Huggingface Transformers library (Wolf et al., 2020). Specifically, it is set to 5 for NC-I2S, NC-S2I, FS, and MG; 3 for NC-I2F and NC-S2F; 1 for all the PP tasks; and 10 for RS. The beam size is set to `num_return_sequences + 3` for all the tasks. The maximum number of new generated tokens is set to 1024.

### D.2 Compared LLMs

We introduce each of the compared LLMs in details, including their training (if applicable) and inference process.

### D.2.1 GPT-4

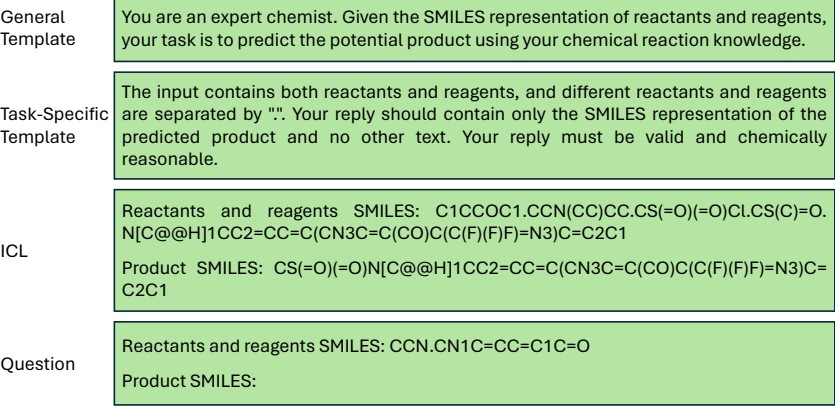

Figure 3: An example of query template for GPT-4.

GPT-4 (OpenAI, 2023) is one of the SoTA LLMs. We use the model versioned as `gpt-4-0613` and evaluate it on 500 samples from SMolInstruct test set via OpenAI's API. Since GPT-4 is not fine-tuned on our dataset and thus is not familiar with the flexible queries, to ensure it generates answers in an expected format, we follow the prompt format proposed in (Guo et al., 2023) and create a query template for each of the tasks. The template for FS is shown in Figure 3. It contains 4 parts: (1) **General template** describes the task in a general way. (2) **Task-specific template** describes the detailed content requirements and format

---

[6] https://huggingface.co/facebook/galactica-6.7b
[7] https://huggingface.co/meta-llama/Llama-2-7b-hf
[8] https://huggingface.co/codellama/CodeLlama-7b-hf
[9] https://huggingface.co/mistralai/Mistral-7B-v0.1

requirements for the specific task. (3) **ICL** contains the in-context learning examples. It provides examples in the format of `<input_title>`: `<input_content>`\n `<output_title>`: `<output_content>`\n, where `<input_title>` and `<output_title>` serve as straightforward prompts to the input and output content. This design make the queried task more clear. (4) **Question** has the same format as ICL, with `<output_content>` being empty for the model to generate.

We conduct both $s$-shot evaluations, where $s = 0, 1, 3, 5$ is the number of provided ICL examples. For 0-shot evaluation, the ICL part in the template is removed from the queries. In $k$-shot evaluation, for each sample, the ICL examples are randomly selected from the training set. The results of these settings are shown in Appendix E, which reals that these settings' performance is not consistent across all the tasks. Since 0-shot shows the best performance on most tasks, we report its results in Section 4.3.

In the evaluations, we use the default generation strategy set in the API. To generate the same number of results for each sample (as described in Appendix D.1), we set the argument n in the API, which controls the number of output sequences.

GPT-4 can always follow the formatted instructions introduced above, so we do not bother to extract the answers from its outputs, but directly use its outputs as the predicted answers.

### D.2.2 Claude 3 Opus

Claude 3 Opus (Anthropic, 2024) is a newly proposed SoTA LLM to date. Similarly to GPT-4, we evaluate Claude 3 Opus on 500 samples from SMolInstruct test set via Anthropic's API, and the generation strategy is the default one. The used prompt format is identical to the one used for GPT-4 (Appendix D.2.1. For each sample, we generate one response. Since Claude 3 Opus can always follow the formatted instructions, we do not bother to extract the answers from its outputs, but directly use its outputs as the predicted answers.

### D.2.3 Galactica

Galactica (Taylor et al., 2022) is a LLM without instruction tuning. To evaluate it on SMolInstruct, we follow the instructions in the paper (Taylor et al., 2022) and the repository[10] to create the queries for each task. We use zero-shot setting, as its official instruction does not suggest using few-shot setting. The generation configuration is set identical to that of our LlaSMol models (Appendix D.1).

Galatica's outputs may contain extra text other than the expected answers. Therefore, with heuristic rules and regular expression matching, we implement a program to extract the answers from the outputs of the models. Since the extraction cannot possibly cover all the possible output formats, some answers might not be correctly extracted, which might lead to validities lower than the actual value.

### D.2.4 Llama 2, Code Llama, and Mistral

For our base models (Llama 2, Code Llama, and mistral), since they are not trained on SMolInstruct and have not seen the diverse queries in the dataset, we use the same query templates as those used for GPT-4 (Appendix D.2.1). We use the one-shot setting for them, as it would improve models' abiltity to follow the instructions and generate answers in a more formated way. In addition, the generation configuration (including beam size, output sequence numbers, etc) is set identical to that of our LlaSMol models (Appendix D.1).

Although we try our best to make the output format as clear as possible in the queries, these three models still cannot follow the instructions and their outputs are in various formats. By heuristic rules and regular expression matching, we implement a program to extract the answers from the outputs of each of the models. Since the extraction cannot possibly cover all the possible output formats, some answers might not be correctly extracted, which might lead to validities lower than the actual value.

---

[10]https://github.com/paperswithcode/galai

### D.2.5 Molinst

Molinst is a Llama 2 model fine-tuned on Mol-Instructions by Fang et al. (2023). On the shared tasks between Mol-Instructions and SMolInstruct (including MC, MG, FS, and RS), we directly use the query templates from Mol-Instructions to achieve better results. On other tasks, we create one query template for each task following the style of Mol-Instructions. We use zero-shot on its evaluation, as Mol-Instructions does not contain any few-shot use cases. We directly use Molinst's checkpoint[11] and evaluate it on SMolInstruct, which is a fair comparison, because SMolInstruct's test set does not contain any training examples from the training set of Mol-Instructions.

The outputs of Molinst may also contain extra text other than the expected answers, especially on its unseen tasks. Thus, we also implement a program to extract the answers. Since the extraction cannot possibly cover all the possible output formats, some answers might not be correctly extracted, which might lead to their validities lower than the actual value.

### D.2.6 ChemLLM

ChemLLM (Zhang et al., 2024) is an LLM simultaneously proposed with ours. We apply their released checkpoint[12] and evaluate it on SMolInstruct. The used prompt format is identical to the one used for GPT-4 (Appendix D.2.1. For each sample, we generate one response.

The outputs of ChemLLM may contain extra text other than the expected answers. Thus, we implement a program to extract the answers from its outputs. Since the extraction cannot possibly cover all the possible output formats, some answers might not be correctly extracted, which might lead to their validities lower than the actual value.

### D.3 Task-Specific, Non-LLM Based SoTA Models

### D.3.1 STOUT for NC-I2S and NC-S2I

STOUT is a encoder-decoder model trained on SMILES and IUPAC name paired data, and it is capable of conducting the NC-2iS and NC-S2I tasks. Due to the lack of training code, we cannot re-train it on our dataset, and directly use their released model checkpoint[13]. Since it may have encounter some test samples of SMolInstruct during training, the evaluation results in Table 1 may be higher than its real performance.

### D.3.2 RDKit for NC-S2F

The NC-S2F task can be easily achieved with a fixed algorithm by parsing the input SMILES representation and counting the numbers of atoms. We implement a program with RDKit (a widely used Python toolkit for processing molecules and other chemical information) and report its results.

### D.3.3 STOUT+RDKit for NC-I2F

Since there are no dedicated models for the NC-I2F task, we combine STOUT for the IUPAC to SMILES conversion and RDKIT for the SMILES to molecular formula conversion. Specifically, we feed the input IUPAC name into STOUT to get the corresponding SMILES, and then used the RDKit-based program to get the molecular formula based on the SMILES.

### D.3.4 Uni-Mol for All The PP Tasks

Uni-Mol (Zhou et al., 2023) is a framework for learning useful representations of molecules based on their 3D conformations. Uni-Mol can be fine-tuned to perform property prediction

---

[11]`https://huggingface.co/zjunlp/llama2-molinst-molecule-7b`.

[12]`https://huggingface.co/AI4Chem/ChemLLM-7B-Chat`

[13]`https://github.com/Kohulan/Smiles-TO-iUpac-Translator`

based on these representations. Using the pretrained model weights, hyperparameters, and code supplied by the authors, we fine-tuned Uni-Mol models for chemical property prediction tasks on our dataset. For SIDER property prediction, we used 20 as the number of targets for multi-target classification, as our dataset focused on a specific subset of 20 SIDER targets. We generated results from Uni-Mol using the code provided by the authors and evaluated according to the metrics in Section 4.2. The data split used for fine-tuning, validation, and testing property prediction tasks is different from the one used in the Uni-Mol paper, so the performance may not match exactly.

### D.3.5 MolT5 for MC and MG

MolT5 (Edwards et al., 2022) is a T5 model for translating between molecules and natural language. We use the already fine-tuned MolT5-large checkpoints provided by the authors for both molecule generation and molecule description. We generate predictions on our test set using beam search with 5 beams, following the example code provided by the authors. For input, the molecule description model is provided a SMILES string and the molecule generation model is provided a natural language description. For molecule description, we generated only one result. For molecule generation, we set the number of sequences to return to 5. We evaluate our test set results according to the metrics in Section 4.2. The data used for testing is different from the data used by the MolT5 paper, so the performance may be different. Please note that our test set does not overlap with the MolT5 training set.

### D.3.6 RSMILES for FS and RS

RSMILES (Zhong et al., 2022) is a transformer model trained on pairs of SMILES strings aligned to minimize their edit distance. RSMILES translates aligned SMILES strings of reactants and reagents into products for the FS task, and products into reactants for the RS task. Following the settings described in the paper of RSMILES, we augment and align each pair of SMILES strings in our training data for 5 times. For the FS task, we adopt a "mixed" setting and append canonical SMILES strings of reagents to the end of aligned reactant SMILES strings. We train two RSMILES models for the FS and RS tasks, respectively, using the hyper-parameters provided in their GitHub repository. After training, we average the last 5 checkpoints to get the final checkpoint for each task. During inference, we augment each input SMILES strings for 5 times. We generate 10 output SMILES strings for each augmented input using beam search, resulting in a total of 50 SMILES strings for each test reaction. We get the final top 10 predictions for each task by aggregating these 50 predictions using their provided scripts.

The performance of our re-trained RSMILES model on our dataset for the RS task is comparable with those reported in their paper on the USPTO-full dataset. Please note that the performance of our re-trained RSMILES for the FS task, as shown in Table 2, is lower than the reported results on the USPTO-MIT dataset for the FS task in their paper. This is due to that our dataset for the FS task is more challenging than the USPTO-MIT dataset used in the RSMILES's paper, due to the inclusion of stereochemical information.

### D.3.7 Molecular Transformer for FS and RS

Similar to RSMILES, Molecular Transformer (Schwaller et al., 2019) is also a transformer model trained on pairs of SMILES strings, and translate from reactants and reagents into products or products into reactants. While the original Molecular Transformer only focused on the FS task, we train and test it on both the FS and RS tasks. We use canonical SMILES strings of molecules without data augmentation as the training data of Molecular Transformer. We train two Molecular Transformer models separately for the FS and RS tasks using the hyper-parameters provided in their GitHub repository. During inference, we generate 10 output SMILES strings for each canonical input SMILES string using beam search. The performance of our re-trained Molecular Transformer model on our dataset for the FS task is comparable with those reported in their paper on the USPTO-STEREO dataset.

### D.4 Evaluation Metrics

We introduce the metrics used in Section 4.3 as follows:

- **Exact match (EM)**. It measures the success of a model in providing responses that perfectly match the reference or ground truth answers. Notably, for each predicted result, we compare it with the gold answers of all the samples that have the same input as the sample. If there exists a match, it is counted as correct and contributes to the ratio of this metric. Note that for different types of outputs, we employ different criterion for judging it they match. For tasks where outputs are SMILES strings (NC-I2S, MG, FS, and RS), we parse SMILES strings into molecules, and they are matched only if the two molecules are identical. For tasks where outputs are molecular formula, two are matched if they represent the same set of atoms, and the corresponding numbers of the samples are identical. For tasks where outputs are IUPAC names (NC-S2I), since IUPAC names may contain multiple parts separated by semicolons, we compare the set composed of these parts. That is, we do not care about the orders of these parts and how many of parts are there in the generated string, but judge by the correctness of the unique parts.
- **Fingerprint Tanimoto Similarity (FTS)**. It is an important metric type commonly used in cheminformatics. It measures the structural similarity between molecules. The one we report in Section 4.3 is one of this type, called Morgan FTS, which leverages Morgan method to calculate the fingerprint (Morgan, 1965).
- **METEOR score**. It is a common metric used to measure the similarity between text. (Lavie & Agarwal, 2007)
- **RMSE**. It is a common metric to measure the distance between predicted values and the gold values on regression tasks. Smaller is better.
- **Acc**. It represents the ratio of correct predictions.
- **Validity (Valid)**: it reports the ratio of valid predicted SMILES representations that can be successfully parsed into a molecule. It is calculated among all the generated outputs that contain extractable answer part. If an output refuses to answer a question, it would not be counted for calculating the validity.

Additional metrics used in Appendix E are briefly introduced as follows:

- **Top-$k$ Exact Match**: It is the same as EM discussed before, but on the top-$k$ generated outputs. It gives a more comprehensive results.
- **MACCS FTS and RDK FTS**: In addition to Morgan FTS we use in the previous sections, we introduce two extra FTS metrics, namely MACCS FTS and RDK FTS, that use MACCS (Durant et al., 2002) and RDK (Schneider et al., 2015) methods to calculate the fingerprint respectively.
- **BLUE scores and ROUGE scores**: Another types of textual based metrics that measures the similarity between text.
- **Matthew's Correlation Coefficient (MCC)**. Applied in the binary classification tasks (PP-BBBP, PP-Clintox, PP-HIV, and PP-SIDER), this metric provides a balanced measure of the quality of binary classifications (Matthews, 1975).
- **F1 Score**: The harmonic mean of precision and recall; a commonly used metric for classification tasks.

## E Detailed Experimental Results

In this section, we present more comprehensive experimental results and analysis. We will provide detailed performance of all models across a wider range of metrics and offer more discussion of the findings. We will discuss about each type of tasks in the rest of this section.

### E.1 Name Conversion Tasks

The results on the four NC tasks are presented in Table 7, Table 8, Table 9, and Table 10. On these tasks, Although open source LLMs including Llama 2, Code Llama, and Mistral can sometimes achieve fairly good validity, showing that they know at least basic knowledge of the molecular representations, they correctly predict none of the samples. It indicates that

their lack of knowledge to do reasoning for the conversions. Trained on scientific corpus (including chemistry materials), Galactica can achieve some correct predictions, but the performance is still low. After fine-tuning on these open source LLMs, LlaSMol models are substantially better than other LLMs, showing the benefits of our dataset and fine-tuning. Compared to the SoTA models, LlaSMol models still underperform, but approach them on NC-I2F, NC-I2S, and NC-S2F. However, on NC-S2I, LlaSMol is still far from SoTA. It seems that NC-S2I is the hardest task among the four NC tasks, as the SoTA task-specific method, STOUT, only achieves 56.5% accuracy. Ability on this task suggests a level of understanding of the functional groups in the IUPAC specification, as well as an understanding of SMILES representations. Improving models' understanding and generation for IUPAC names might be necessary.

Table 7: Overall results (%) of NC-I2F.

| Model | EM | | Validity |
|---|---|---|---|
| | Top 1 | Top 3 | |
| STOUT+RDKit | 97.9 | - | 100.0 |
| GPT-4 (0-shot) | 8.7 | 16.4 | 98.4 |
| GPT-4 (1-shot) | 10.8 | 19.9 | 98.6 |
| GPT-4 (3-shot) | 9.1 | 17.7 | 98.6 |
| GPT-4 (5-shot) | 11.5 | 20.4 | 98.8 |
| Claude 3 Opus | 34.6 | - | 98.2 |
| Galactica | 9.1 | 12.3 | **100.0** |
| Llama 2 | 0.0 | 0.1 | 96.7 |
| Code Llama | 0.0 | 0.0 | 98.5 |
| Mistral | 0.0 | 0.2 | 98.5 |
| Molinst | 0.0 | 0.0 | 1.7 |
| ChemLLM | 0.8 | - | 97.4 |
| LlaSMol$_{\text{Galactica}}$ | 83.2 | 91.3 | **100.0** |
| LlaSMol$_{\text{Llama 2}}$ | 73.8 | 86.1 | **100.0** |
| LlaSMol$_{\text{Code Llama}}$ | 75.4 | 87.0 | **100.0** |
| LlaSMol$_{\text{Mistral}}$ | **87.9** | **93.2** | **100.0** |

Table 8: Overall results (%) of NC-I2S.

| Model | Exact Match | | | FTS | | | Validity |
|---|---|---|---|---|---|---|---|
| | Top 1 | Top 3 | Top 5 | MACCS | RDK | Morgan | |
| STOUT | 73.5 | - | - | 99.9 | 99.8 | 99.5 | 99.4 |
| GPT-4 (0-shot) | 3.3 | 5.1 | 5.9 | 77.6 | 52.0 | 49.4 | 84.2 |
| GPT-4 (1-shot) | 3.3 | 5.7 | 6.9 | 76.5 | 49.6 | 48.1 | 85.8 |
| GPT-4 (3-shot) | 3.6 | 5.9 | 6.9 | 76.5 | 48.8 | 46.9 | 84.4 |
| GPT-4 (5-shot) | 2.4 | 4.7 | 6.1 | 75.6 | 47.5 | 46.2 | 84.8 |
| Claude 3 Opus | 17.7 | - | - | 88.5 | 70.0 | 68.6 | 90.2 |
| Galactica | 9.7 | 11.1 | 12.5 | 81.5 | 58.1 | 53.4 | 95.6 |
| Llama 2 | 0.0 | 0.0 | 0.0 | 29.4 | 18.7 | 11.3 | 18.3 |
| Code Llama | 0.0 | 0.0 | 0.0 | 30.7 | 20.0 | 12.0 | 81.0 |
| Mistral | 0.0 | 0.0 | 0.0 | 33.6 | 21.3 | 11.3 | 40.3 |
| Molinst | 0.0 | 0.0 | 0.0 | 43.9 | 25.1 | 18.4 | 96.2 |
| ChemLLM | 0.3 | - | - | 0.0 | 0.0 | 0.0 | 3.9 |
| LlaSMol$_{\text{Galactica}}$ | 58.7 | 68.8 | 72.6 | 95.5 | 86.4 | 84.9 | 99.4 |
| LlaSMol$_{\text{Llama 2}}$ | 46.6 | 57.5 | 60.5 | 92.4 | 79.3 | 78.3 | 99.0 |
| LlaSMol$_{\text{Code Llama}}$ | 49.9 | 60.1 | 63.8 | 93.1 | 80.9 | 80.0 | 99.3 |
| LlaSMol$_{\text{Mistral}}$ | **70.1** | **77.8** | **80.1** | **96.6** | **90.1** | **89.1** | **99.6** |

Table 9: Overall results (%) of NC-S2F.

| Model | EM | | Validity |
| --- | --- | --- | --- |
| | Top 1 | Top 3 | |
| RDKit | 100.0 | - | 100.0 |
| GPT-4 (0-shot) | 4.8 | 11.6 | 99.8 |
| GPT-4 (1-shot) | 3.4 | 9.2 | **100.0** |
| GPT-4 (3-shot) | 3.4 | 9.0 | 99.8 |
| GPT-4 (5-shot) | 3.0 | 8.0 | 99.4 |
| Claude Opus 3 | 9.2 | - | **100.0** |
| Galactica | 0.0 | 0.0 | 99.9 |
| Llama 2 | 0.0 | 0.0 | 72.1 |
| Code Llama | 0.0 | 0.0 | 97.1 |
| Mistral | 0.0 | 0.1 | 87.8 |
| Molinst | 0.0 | 0.0 | 19.3 |
| ChemLLM | 0.0 | - | 25.8 |
| LlaSMol$_{\text{Galactica}}$ | 91.2 | 95.0 | **100.0** |
| LlaSMol$_{\text{Llama 2}}$ | 87.0 | 93.1 | **100.0** |
| LlaSMol$_{\text{Code Llama}}$ | 88.6 | 94.7 | **100.0** |
| LlaSMol$_{\text{Mistral}}$ | **93.2** | **96.5** | **100.0** |

Table 10: Overall results (%) of NC-S2I.

| Model | EM | | |
| --- | --- | --- | --- |
| | Top 1 | Top 3 | Top 5 |
| STOUT | 56.5 | - | - |
| GPT-4 (0-shot) | 0.0 | 0.0 | 0.0 |
| GPT-4 (1-shot) | 0.0 | 0.0 | 0.0 |
| GPT-4 (3-shot) | 0.2 | 0.2 | 0.2 |
| GPT-4 (5-shot) | 0.2 | 0.2 | 0.2 |
| Claude 3 Opus | 0.0 | - | - |
| Galactica | 0.0 | 0.0 | 0.0 |
| Llama 2 | 0.0 | 0.0 | 0.0 |
| Code Llama | 0.0 | 0.0 | 0.0 |
| Mistral | 0.0 | 0.0 | 0.0 |
| Molinst | 0.0 | 0.0 | 0.0 |
| ChemLLM | 0.0 | - | - |
| LlaSMol$_{\text{Galactica}}$ | 18.3 | 31.8 | 36.8 |
| LlaSMol$_{\text{Llama 2}}$ | 12.9 | 23.2 | 26.6 |
| LlaSMol$_{\text{Code Llama}}$ | 15.5 | 26.2 | 30.5 |
| LlaSMol$_{\text{Mistral}}$ | **29.0** | **45.3** | **50.5** |

### E.2 Property Prediction

Table 11: Overall results (RMSE) of PP-ESOL and PP-Lipo.

| Model | ESOL↓ | Lipo↓ |
|---|---|---|
| Uni-Mol | 0.819 | 0.612 |
| GPT-4 (0-shot) | 2.570 | 1.545 |
| GPT-4 (1-shot) | 2.268 | 1.625 |
| GPT-4 (3-shot) | 2.027 | 1.777 |
| GPT-4 (5-shot) | 1.689 | 1.592 |
| Claude 3 | **1.036** | 1.194 |
| Galactica | 4.184 | 2.979 |
| Llama 2 | 3.287 | 1.634 |
| Code Llama | 3.483 | 1.733 |
| Mistral | 3.079 | 1.730 |
| Molinst | 4.304 | 2.800 |
| ChemLLM | 6.635 | 2.499 |
| LlaSMol$_{Galactica}$ | 1.959 | 1.213 |
| LlaSMol$_{Llama 2}$ | 2.791 | 1.338 |
| LlaSMol$_{Code Llama}$ | 2.959 | 1.203 |
| LlaSMol$_{Mistral}$ | 1.150 | **1.010** |

The results on the two regression tasks (PP-ESOL and PP-Lipo) are presented in Table 11. LlaSMol$_{Mistral}$ outperforms other LLMs including GPT-4 by a large margin, showing the effectiveness of the fine-tuning. However, it is slightly worse than Claude 3 Opus on PP-ESOL, possibly because of its much larger scale of parameters and training data. The task-specific method (Uni-Mol) still outperforms all of the LLMs. This might be attributed to the fact that Uni-Mol (Zhou et al., 2023) possesses 3D structure knowledge, which may be useful in predicting molecular properties.

Table 12: Overall results (Acc) of PP-BBBP, PP-ClinTox, PP-HIV, and PP-SIDER.

| Model | BBBP | ClinTox | HIV | SIDER |
|---|---|---|---|---|
| Uni-Mol | 85.3 | 92.4 | 97.0 | 70.0 |
| GPT-4 (0-shot) | 62.9 | 50.0 | 59.6 | 57.6 |
| GPT-4 (1-shot) | 66.0 | 25.7 | 39.4 | 43.2 |
| GPT-4 (3-shot) | 60.9 | 36.1 | 48.0 | 39.8 |
| GPT-4 (5-shot) | 57.9 | 33.3 | 55.8 | 50.4 |
| Claude 3 Opus | **75.1** | 41.7 | 76.4 | 67.0 |
| Galactica | 69.0 | 92.4 | **96.7** | 68.1 |
| Llama 2 | 58.9 | 45.1 | 93.3 | 61.9 |
| Code Llama | 58.9 | 85.4 | 91.8 | 60.2 |
| Mistral | 40.6 | 15.3 | 7.1 | 38.1 |
| Molinst | 60.9 | 6.3 | 4.5 | 52.4 |
| ChemLLM | 22.3 | 75.7 | 72.9 | 32.6 |
| LlaSMol$_{Galactica}$ | 69.0 | 93.1 | **96.7** | 70.1 |
| LlaSMol$_{Llama 2}$ | 69.0 | 92.4 | **96.7** | 68.7 |
| LlaSMol$_{Code Llama}$ | 69.0 | 93.1 | **96.7** | 69.9 |
| LlaSMol$_{Mistral}$ | 74.6 | **93.1** | **96.7** | **70.7** |

The results for the four classification tasks (PP-BBBP, PP-Clintox, PP-HIV, and PP-SIDER) presented in (Table 12) show similar information. Particularly, on PP-SIDER task, LlaSMol$_{Mistral}$ outperforms all the LLMs and the task-specific model Uni-Mol, which highlights the potential of LLMs in understanding molecules and predicting their properties. The reason why

LlaSMol models get the same results on PP-HIV is that the dataset is imbalanced, and the models all predict most samples as negative samples.

### E.3 Molecule Description

For the MC task (Table 13), LlaSMol$_\text{Mistral}$ is the best performing LLM on all metrics. It is still outperformed by the SoTA task-specific model (MolT5), however, approaching close to it. Please note that these text-based metrics only measures the similarity to the gold descriptions, and it does not necessary mean the correctness of the description on chemistry dimension. Limited by the current resources, we cannot obtain the correctness measures, and will leave this to our future work.

Table 13: Overall results of MC.

| Model | BLEU-2 | BLEU-4 | ROUGE-1 | ROUGE-2 | ROUGE-L | METEOR |
|---|---|---|---|---|---|---|
| MolT5 | 0.462 | 0.366 | 0.563 | 0.398 | 0.501 | 0.515 |
| GPT-4 (0-shot) | 0.095 | 0.020 | 0.238 | 0.058 | 0.156 | 0.188 |
| GPT-4 (1-shot) | 0.166 | 0.061 | 0.295 | 0.099 | 0.211 | 0.206 |
| GPT-4 (3-shot) | 0.202 | 0.092 | 0.333 | 0.132 | 0.250 | 0.244 |
| GPT-4 (5-shot) | 0.214 | 0.103 | 0.346 | 0.146 | 0.267 | 0.258 |
| Claude 3 Opus | 0.114 | 0.030 | 0.263 | 0.069 | 0.173 | 0.219 |
| Galactica | 0.018 | 0.002 | 0.061 | 0.012 | 0.052 | 0.050 |
| Llama 2 | 0.110 | 0.047 | 0.251 | 0.107 | 0.209 | 0.150 |
| Code Llama | 0.106 | 0.052 | 0.247 | 0.122 | 0.216 | 0.143 |
| Mistral | 0.146 | 0.068 | 0.281 | 0.118 | 0.232 | 0.193 |
| Molinst | 0.028 | 0.020 | 0.226 | 0.160 | 0.217 | 0.124 |
| ChemLLM | 0.012 | 0.004 | 0.057 | 0.005 | 0.048 | 0.050 |
| LlaSMol$_\text{Galactica}$ | 0.314 | 0.225 | 0.456 | 0.289 | 0.402 | 0.394 |
| LlaSMol$_\text{Llama 2}$ | 0.302 | 0.213 | 0.447 | 0.281 | 0.396 | 0.377 |
| LlaSMol$_\text{Code Llama}$ | 0.322 | 0.227 | 0.441 | 0.273 | 0.390 | 0.366 |
| LlaSMol$_\text{Mistral}$ | **0.414** | **0.319** | **0.521** | **0.357** | **0.463** | **0.452** |

For the MG task (Table 14), LlaSMol$_\text{Mistral}$ is the best performing LLM on all metrics. Since this task aims to generate an arbitrary molecule that fits the given description, it is not required to exactly match the gold one, and the FTS metrics may be better to reflect models' ability. However, similar to MC, the metrics only measure the similarities between predicted and gold ones, and not necessarily means the correctness of the generation. We will leave this to our future work.

### E.4 Chemical Reaction

For the FS task (Table 15), LlaSMol$_\text{Mistral}$ is the best performing LLM across all metrics, although it ties with many methods on validity. Notably, all of the LlaSMol models perform much better than the other LLMs, which indicates the power of fine-tuning on SMolInstruct for understanding chemical reactions. The SoTA task-specific methods still outperform all of the LLMs, but the LlaSMol series is much closer than the other LLMs.

We observe a similar trend for the RS task (Table 16). Again, LlaSMol$_\text{Mistral}$ is the best performing LLM across all metrics, although it does tie with LlaSMol$_\text{Llama 2}$ for validity. We observe the LLMs without instruction tuning fail to achieve any accuracy greater than 2% on this task. This indicates that instruction tuning can be useful for LLMs to learn retrosynthesis. The SoTA task-specific methods still outperform all of the LLMs, which indicates that there is still room for improvement for LLMs on RS.

### E.5 Other Common Findings

Besides for the aforementioned findings, we can observe some common findings across different tasks. Firstly, we can see that GPT-4 does not show consistent pattern on different

Table 14: Overall results (%) of MG.

| Model | Exact Match | | | FTS | | | Validity |
|---|---|---|---|---|---|---|---|
| | Top 1 | Top 3 | Top 5 | MACCS | RDK | Morgan | |
| MolT5 | 31.7 | 38.7 | 41.4 | 87.9 | 80.2 | 73.2 | 95.3 |
| GPT-4 (0-shot) | 6.4 | 7.7 | 9.2 | 74.2 | 53.3 | 42.6 | 81.4 |
| GPT-4 (1-shot) | 4.9 | 6.2 | 7.9 | 74.0 | 52.9 | 42.8 | 81.8 |
| GPT-4 (3-shot) | 5.9 | 8.2 | 8.9 | 74.8 | 53.5 | 43.3 | 85.2 |
| GPT-4 (5-shot) | 4.0 | 5.9 | 7.1 | 73.6 | 52.8 | 43.1 | 85.2 |
| Claude 3 Opus | 12.3 | - | - | 83.9 | 67.6 | 57.6 | 92.6 |
| Galactica | 0.0 | 0.0 | 0.0 | 22.7 | 11.8 | 11.6 | 94.7 |
| Llama 2 | 0.0 | 0.0 | 0.0 | 18.3 | 11.8 | 4.8 | 93.5 |
| Code Llama | 0.0 | 0.0 | 0.0 | 26.5 | 15.1 | 8.5 | 95.2 |
| Mistral | 0.0 | 0.1 | 0.1 | 32.2 | 18.4 | 9.0 | 35.9 |
| Molinst | 6.0 | 11.7 | 13.4 | 69.5 | 53.5 | 43.6 | 84.8 |
| ChemLLM | 0.9 | - | - | 38.3 | 21.5 | 14.3 | 4.3 |
| LlaSMol$_{\text{Galactica}}$ | 7.7 | 14.2 | 17.4 | 79.6 | 62.0 | 52.2 | 99.6 |
| LlaSMol$_{\text{Llama 2}}$ | 6.4 | 11.4 | 13.7 | 74.3 | 55.8 | 47.1 | 99.6 |
| LlaSMol$_{\text{Code Llama}}$ | 6.5 | 11.8 | 14.2 | 74.0 | 55.9 | 46.6 | **99.7** |
| LlaSMol$_{\text{Mistral}}$ | **19.2** | **29.0** | **33.6** | **84.1** | **70.4** | **61.7** | **99.7** |

Table 15: Overall results (%) of FS.

| Model | Exact Match | | | FTS | | | Validity |
|---|---|---|---|---|---|---|---|
| | Top 1 | Top 3 | Top 5 | MACCS | RDK | Morgan | |
| Molecular Transformer | 78.4 | 85.5 | 87.0 | 95.5 | 92.9 | 91.4 | 99.5 |
| RSMILES | 78.7 | 88.0 | 89.7 | 95.7 | 93.7 | 92.2 | 100.0 |
| GPT-4 (0-shot) | 1.6 | 2.4 | 2.6 | 60.8 | 49.4 | 40.5 | 87.0 |
| GPT-4 (1-shot) | 1.1 | 2.2 | 2.6 | 61.5 | 49.5 | 41.1 | 91.4 |
| GPT-4 (3-shot) | 0.2 | 2.2 | 2.6 | 62.2 | 51.3 | 42.8 | 92.0 |
| GPT-4 (5-shot) | 1.3 | 2.0 | 3.0 | 63.1 | 51.7 | 44.0 | 93.8 |
| Claude 3 Opus | 3.7 | - | - | 65.7 | 54.1 | 45.7 | 97.0 |
| Galactica | 0.0 | 0.0 | 0.0 | 40.2 | 33.2 | 25.9 | 83.7 |
| Llama 2 | 0.0 | 0.0 | 0.0 | 33.4 | 24.3 | 13.7 | 97.7 |
| Code Llama | 0.0 | 0.0 | 0.0 | 35.3 | 26.4 | 15.8 | 99.6 |
| Mistral | 0.0 | 0.0 | 0.0 | 38.9 | 31.0 | 19.9 | 95.8 |
| Molinst | 2.1 | 3.3 | 3.7 | 51.1 | 36.7 | 31.7 | **99.8** |
| ChemLLM | 0.0 | - | - | 8.0 | 0.9 | 1.6 | 38.5 |
| LlaSMol$_{\text{Galactica}}$ | 53.1 | 66.4 | 70.7 | 88.8 | 82.6 | 79.9 | 99.7 |
| LlaSMol$_{\text{Llama 2}}$ | 47.1 | 61.6 | 66.4 | 87.0 | 80.1 | 76.9 | **99.8** |
| LlaSMol$_{\text{Code Llama}}$ | 52.0 | 65.4 | 69.2 | 88.3 | 81.9 | 79.2 | **99.8** |
| LlaSMol$_{\text{Mistral}}$ | **63.3** | **75.5** | **79.0** | **91.8** | **87.1** | **84.9** | **99.8** |

Table 16: Overall results (%) of RS.

| Model | Exact Match | | | FTS | | | Validity |
|---|---|---|---|---|---|---|---|
| | Top 1 | Top 3 | Top 5 | MACCS | RDK | Morgan | |
| Molecular Transformer | 47.0 | 61.7 | 66.5 | 87.0 | 81.5 | 77.5 | 99.7 |
| RSMILES | 46.2 | 63.9 | 69.9 | 86.5 | 81.0 | 76.8 | 100.0 |
| GPT-4 (0-shot) | 0.0 | 0.3 | 0.2 | 57.5 | 34.0 | 33.4 | 42.6 |
| GPT-4 (1-shot) | 0.3 | 0.8 | 1.4 | 66.6 | 42.5 | 40.9 | 79.6 |
| GPT-4 (3-shot) | 0.5 | 1.2 | 1.6 | 68.2 | 45.6 | 42.2 | 87.8 |
| GPT-4 (5-shot) | 0.2 | 0.8 | 1.2 | 68.3 | 46.0 | 43.1 | 84.4 |
| Claude 3 Opus | 1.1 | - | - | 70.3 | 49.8 | 46.2 | 94.8 |
| Galactica | 0.0 | 0.0 | 0.0 | 48.9 | 38.2 | 34.6 | 93.0 |
| Llama 2 | 0.0 | 0.0 | 0.0 | 46.6 | 35.0 | 27.5 | 87.7 |
| Code Llama | 0.0 | 0.1 | 0.0 | 44.7 | 32.1 | 25.3 | 97.1 |
| Mistral | 0.0 | 0.0 | 0.0 | 44.6 | 32.0 | 24.2 | 98.0 |
| Molinst | 5.7 | 8.3 | 9.5 | 69.6 | 53.7 | 48.0 | 97.8 |
| ChemLLM | 0.0 | - | - | 14.3 | 2.1 | 2.9 | 10.9 |
| LlaSMol$_{\text{Galactica}}$ | 25.7 | 40.8 | 46.3 | 80.8 | 71.9 | 67.0 | 99.9 |
| LlaSMol$_{\text{Llama 2}}$ | 22.5 | 35.5 | 41.1 | 79.8 | 70.4 | 65.2 | 99.9 |
| LlaSMol$_{\text{Code Llama}}$ | 25.7 | 40.0 | 45.7 | 80.7 | 71.7 | 66.7 | **100.0** |
| LlaSMol$_{\text{Mistral}}$ | **32.9** | **49.6** | **55.4** | **83.0** | **75.2** | **70.4** | **100.0** |

in-context learning settings, with 0-shot achieving the best performance on most tasks. It may indicate that GPT-4 does not have sufficient knowledge about chemistry and cannot effectively learn how to do these tasks by imitating several samples. Injecting chemistry knowledge by training LLMs may be necessary. Secondly, although Claude 3 Opus still trails behind LlaSMol and the SoTA task-specific models, it surpasses GPT-4 on all the chemistry tasks. Claude 3 Opus demonstrates notably enhanced performance compared to GPT-4, showcasing its superior grasp of chemistry knowledge, highlighting its potential applications in the domain.

# F  More Analytical Experiments

## F.1  Task Synergy

In order to investigate the synergy among different tasks, we first conduct an experiment to evaluate the performance of an LLM trained on single task or a specific type of task. We train several single-task models, each focusing on one task or a group of tasks from the same type[14]. We then compare the performance of these single-task models with LlaSMol$_{\text{Mistral}}$, which is trained on all 14 tasks, to assess the potential benefits of multi-task training.

The results are presented in Table 17. The multi-task trained model, LlaSMol$_{\text{Mistral}}$, outperforms the single-task models on the majority of tasks. The improvements are particularly substantial on PP-ESOL, PP-BBBP, PP-Clintox, MC, and MG. This finding suggests the presence of shared knowledge across these tasks. For instance, the knowledge of understanding SMILES may be common to most tasks, and training models on multiple tasks can enhance this knowledge, thus achieving better performance.

To further investigate the relationships among different tasks, we conduct an additional experiment by removing certain tasks from the training data. Specifically, we train a set of models, each with one task or a group of tasks from the same type removed from its training

---

[14]These single-task models include LlaSMol-NC, LlaSMol-PP, LlaSMol-MC, LlaSMol-MG, LlaSMol-FS, and LlaSMol-RS, where the model names indicate the training task(s). For instance, LlaSMol-FS is trained only on the FS task, LlaSMol-NC is trained on all the NC tasks combined, and LlaSMol-PP is trained on all the PP tasks combined. The training setups are identical to those of LlaSMol$_{\text{Mistral}}$.

Table 17: Results of single-task models and multi-task models. The "Single-Task" column corresponds to the single-task models trained on the corresponding tasks, while the "Multi-Task" column corresponds to LlaSMol$_{\text{Mistral}}$ that is trained on all the tasks. Orange cells represent positive performance improvement.

| Task | Metric | Single-Task | Multi-Task | Improv. |
|---|---|---|---|---|
| NC-I2F | EM (%) | 86.8 | 87.9 | 1.1 |
| NC-I2S | EM (%) | 67.6 | 70.1 | 2.4 |
| NC-S2F | EM (%) | 93.2 | 93.2 | 0.0 |
| NC-S2I | EM (%) | 27.4 | 29.0 | 1.5 |
| PP-ESOL | RMSE↓ | 20.616 | 1.150 | 19.466 |
| PP-Lipo | RMSE↓ | 1.241 | 1.010 | 0.231 |
| PP-BBBP | Acc (%) | 68.5 | 74.6 | 6.1 |
| PP-Clintox | Acc (%) | 79.9 | 93.1 | 13.2 |
| PP-HIV | Acc (%) | 96.7 | 96.7 | 0.0 |
| PP-SIDER | Acc (%) | 64.3 | 70.7 | 6.4 |
| MC | METEOR | 0.299 | 0.452 | 0.153 |
| MG | FTS (%) | 33.1 | 61.7 | 28.6 |
| FS | EM (%) | 62.6 | 63.3 | 0.7 |
| RS | EM (%) | 31.5 | 32.9 | 1.4 |

data[15], and their base models and training setups are identical to those of LlaSMol$_{\text{Mistral}}$. We then compare their performance with LlaSMol$_{\text{Mistral}}$, which is trained without removing any tasks.

Table 18: Results of removing certain tasks. Orange cells represent better results than LlaSMol$_{\text{Mistral}}$ while blue cells represent worse results.

| Model | NC-I2F EM (%) | NC-I2S EM (%) | NC-S2F EM (%) | NC-S2I EM (%) | PP-ESOL RMSE | PP-Lipo RMSE | PP-BBBP Acc (%) | PP-Clintox Acc (%) | PP-HIV Acc (%) | PP-SIDER Acc (%) | MC METEOR | MG FTS (%) | FS EM (%) | RS EM (%) |
|---|---|---|---|---|---|---|---|---|---|---|---|---|---|---|
| w/o NC | - | - | - | - | 1.520 | 1.090 | 76.1 | 93.1 | 96.8 | 70.6 | 0.436 | 54.9 | 63.2 | 33.5 |
| w/o PP | 87.9 | 70.7 | 93.5 | 28.7 | - | - | - | - | - | - | 0.447 | 62.3 | 64.2 | 33.1 |
| w/o MC | 87.6 | 71.0 | 93.5 | 27.8 | 1.133 | 1.057 | 74.1 | 93.1 | 96.8 | 70.9 | - | 64.1 | 63.3 | 33.4 |
| w/o MG | 87.8 | 69.6 | 93.4 | 27.8 | 1.231 | 0.982 | 77.2 | 93.1 | 96.8 | 70.9 | 0.445 | - | 63.4 | 34.0 |
| w/o FS | 87.9 | 70.4 | 93.8 | 29.5 | 1.278 | 1.288 | 70.6 | 93.1 | 96.8 | 70.8 | 0.452 | 63.2 | - | 33.1 |
| w/o RS | 88.0 | 71.1 | 93.7 | 29.7 | 1.203 | 1.048 | 72.1 | 93.1 | 96.8 | 70.6 | 0.450 | 62.6 | 61.9 | - |
| LlaSMol$_{\text{Llama 2}}$ | 87.9 | 70.1 | 93.2 | 29.0 | 1.150 | 1.010 | 74.6 | 93.1 | 96.7 | 70.7 | 0.452 | 61.7 | 63.3 | 32.9 |

Interestingly, results presented in Table 18 generally show no consistent pattern among different tasks. Specifically, removing certain task(s) can lead to improvements on some tasks, while leading to decrements on others. And in most cases, the change is not substantial, suggesting that each task does not heavily rely on any other tasks, but rather on the data of the task itself. However, there are two exceptions – for the "w/o NC" model, the performance of PP-ESOL and MG greatly drops. On PP-ESOL, we hypothesize that the knowledge of understanding chemical structures learned on NC might be useful in predicting the solubility of a molecule. For MG, since there are many IUPAC names in the input molecular descriptions, the performance drop may be attributed to the fact that the ability to convert IUPAC names to SMILES, learned on NC, can help the model directly obtain the related SMILES representations, thus leading to better performance.

### F.2 Influence of LoRA Modules and Trainable Parameters

In this section, we investigate the influence of using different LoRA modules, or different sizes of trainable parameters. We take LlaSMol$_{\text{Mistral}}$ as the basic setting and refer to it as LlaSMol in this section for simplicity. All the compared models are listed as follows, with trainable parameter sizes and ratios labeled in brackets:

- **LlaSMol Lite** (6.8M, 0.09%): LoRA is applied on `q_proj` and `v_proj` of the attention modules.
- **LlaSMol Attn** (13.6M, 0.19%): LoRA is applied on all the attention projection matrices (including `q_proj`, `k_proj`, `v_proj`, `o_proj`).

---

[15]These models are named as "w/o Task", where Task represents the task names.

- **LlaSMol FFN** (28.3M, 0.39%): LoRA is applied on all the FFN projection matrices (including `gate_proj`, `down_proj`, `up_proj`).
- **LlaSMol** (41.9M, 0.58%): The basic setting. LoRA is applied on all the attention and FFN projection matrices.
- **LlaSMol Plus** (173.0M, 2.33%): LoRA is applied on all the attention and FFN projection matrices, and `lm_head` is set trainable.

All these models are trained with the identical training configurations (as described in Section 4.1).

Figure 4 presents the model performance on all the 14 tasks. We can observe that for most tasks, progressing from LlaSMol Lite to LlaSMol Attn, LlaSMol FFN, and finally LlaSMol, the incorporation of more LoRA modules (and thus more trainable parameters) leads to a substantial performance enhancement. Further adding `lm_head` as trainable parameters in LlaSMol Plus can further slightly improves the performance. This indicates that refining the selection of LoRA modules and incorporating more trainable parameters is very important. Moreover, LLMs exhibit considerable potential to surpass the performance of previous task-specific models on various chemistry tasks, provided that more trainable parameters are allowed or even full fine-tuning is employed.

However, for certain property prediction (PP) tasks, the incorporation of additional LoRA modules does not demonstrate consistent improvements. This inconsistency may be attributed to the relatively small number of training and test samples, which can result in reduced model robustness when faced with diverse inputs, as well as the inherent randomness associated with limited data.

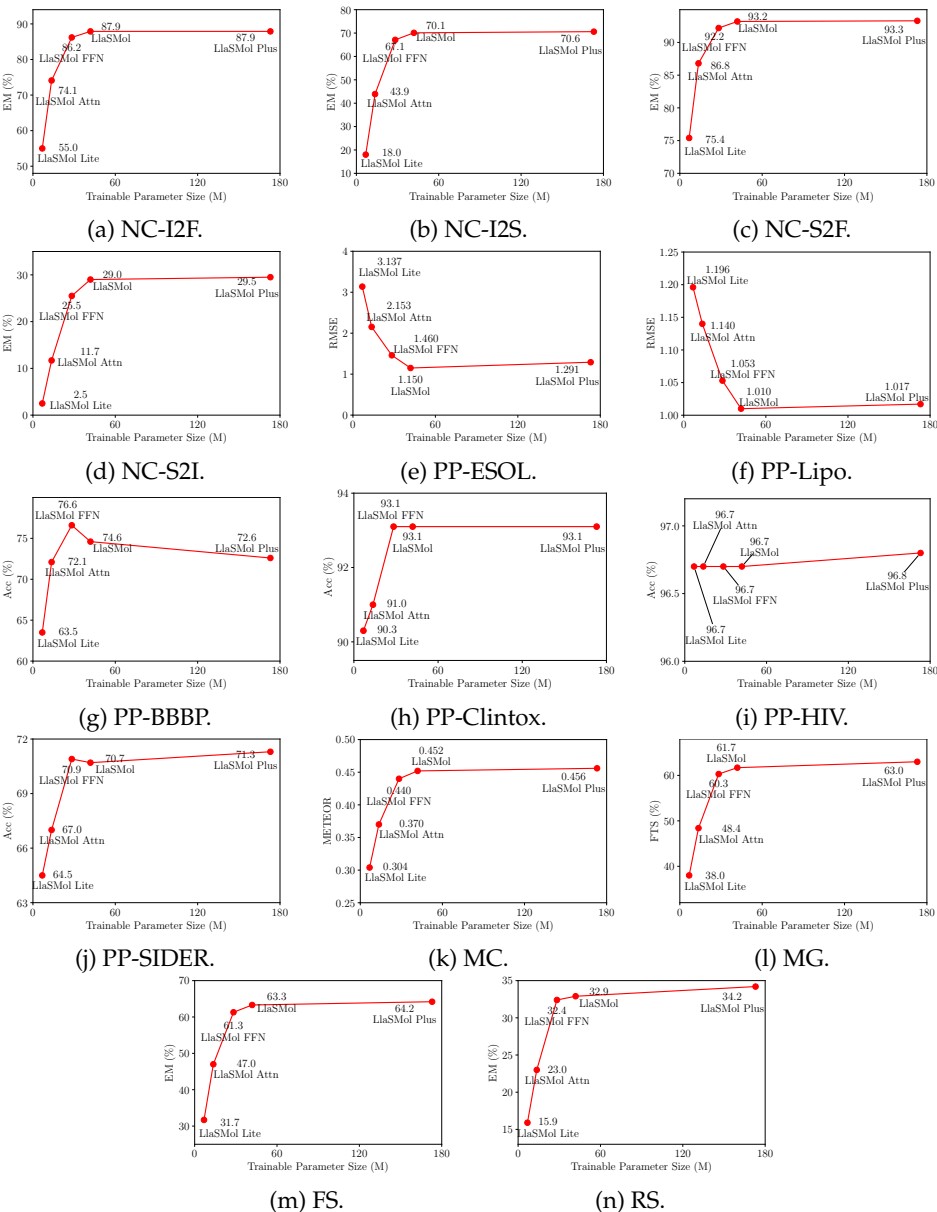

Figure 4: Performance on different tasks under different LoRA settings. Except for PP-ESOL and PP-Lipo, the larger the better.

