# OpenReview forum: "LlaSMol: Advancing Large Language Models for Chemistry with a Large-Scale, Comprehensive, High-Quality Instruction Tuning Dataset"
_colmweb.org/COLM/2024/Conference — COLM_

### Official Review · Reviewer_fwDB · 2024-05-03

**Rating:** 7
**Confidence:** 4
**Ethics Flag:** 1

**Summary:**

This paper proposes a new instruction-tuning dataset, SMolInstruct, that includes 3M+ data points across 14 chemistry tasks, divided into 4 main categories, to help improve the performance of large language models in chemistry. To this end, the paper describes the data collection, curation, and splitting processes used to obtain SMolInstruct, and empirically evaluates the performance of standard open LLMs fine-tuned on this dataset using LoRA against both (i) state-of-the-art (SoTA) models that narrowly focus on individual tasks, and (ii) state-of-the-art language models like GPT-4 and Claude 3. The paper finds that fine-tuning on SMolInstruct allows the base LLM to substantially outperform the best-performing LLMs and narrow the gap with the SoTA models for the individual tasks significantly. To support these findings, the paper also conducts ablation studies by changing the underlying molecular representation and fine-tuning on different instruction tuning dataset, observing that SMolInstruct is the key factor behind the observed empirical improvements.

**Questions To Authors:**

- Why did you not follow existing benchmarks (where available) for the tasks at hand?

**Reasons To Accept:**

- The proposed dataset is large, well-curated, and a useful addition to the literature.
- The empirical results shown in the paper are encouraging. Moreover, they are obtained with little computational resources, and thus there is strong reason to believe that a more rigorous computational exploration could unlock further improvements.
- The discussion on underlying molecular representations and the importance of the fine-tuning process is insightful and important. Indeed, the argument about pre-performing canonicalization highlights that LLMs inherently struggle with more structure-related tasks, and further  support the need for better-designed datasets, as well as more careful integration of LLMs with specialized components.
- The empirical analysis (baselines, protocol) is comprehensive and insightful.

**Reasons To Reject:**

- The paper's argument about the differences in the fine-tuned base model performances is not rigorous enough. In particular, the hyper-parameter tuning conducted for these results is not complete enough to conclusively assert the performance differences. The point raised about the original training is plausible and pertinent. However, I would still tone this down to better reflect the significance of the results.

- The paper does not evaluate on existing benchmarks for the tasks being studied, instead proposing its own test split on its own data, which makes findings less comparable with other approaches. There are well-established benchmarks for at least some of these tasks. For instance, molecular property prediction benchmarks based on MoleculeNet are very common, e.g., in the area of graph neural networks (GNNs), which the authors cite (see the OGB [1] paper for the specific benchmarks). Hence, I strongly advise the authors to re-design their splits to align with existing baselines, i.e., vetting the training split of SMolInstruct, which includes MoleculeNet data, to avoid leakage w.r.t, the established baselines,  and then running similar experiments on the known benchmarks. This would not only provide a real-world reference task with relevant baselines, but also consolidate the generality of your results.


[1] Hu et al. Open Graph Benchmark: Datasets for Machine Learning on Graphs, NeurIPS 2020.

---

> ### Author Rebuttal · Authors · 2024-05-31
>
> Thank you for characterizing our work as “large, well-curated, useful dataset”, “the encouraging empirical results”, “insightful discussion”, and “comprehensive and insightful empirical analysis”. We address your concerns and questions below.
>
> ---
> **“hyper-parameter tuning is not complete enough”**
>
> Thanks for pointing it out. We acknowledge that, due to computational resource constraints, we adopted a limited search space, which might lead to suboptimal performance for some models. We will follow your suggestions and tone down the differences in the fine-tuned model performances in our updated draft.
>
> ---
> **“not evaluate on existing benchmarks for the tasks being studied” “not follow existing benchmarks (where available) for the tasks at hand”**
>
> Thank you for your insightful concern. To ensure rigor and comparability with existing work, we have dedicated much effort in aligning our dataset with existing benchmarks, as detailed in Section 3.2. Specifically:
> - We matched our data splitting for shared tasks (MC, MG, FS, and RS) with the Mol-Instructions dataset [1]. This means the SMolInstruct test set excludes any training examples from Mol-Instructions, allowing us to directly evaluate models trained on Mol-Instructions (Molinst in Tables 1 and 2).
> - For molecular property prediction, our data comes exactly from MoleculeNet. Since it does not provide a specific sample splitting, we had to do the splitting ourselves. To ensure fair comparisons, we re-trained the compared models on our splits using their official codebases. Thus, we have ensured all comparisons in our manuscript are fair by either aligning data splits with existing models’ training data or retraining models on our splits.
>
> Our primary focus is performance on our dataset, so we did not train and evaluate models on other benchmarks. To compare our dataset with others, we trained LlaSMol on Mol-Instructions and evaluated it on our SMolInstruct. As shown in Table 4, even on shared tasks, the model trained on SMolInstruct outperforms the model trained on Mol-Instructions, highlighting the advantages of SMolInstruct.
>
> [1] Fang, Y., et. al. (2023). Mol-Instructions: A Large-Scale Biomolecular Instruction Dataset for Large Language Models. In ICLR.
>
> ---
> Thank you again for your constructive feedback. We will revise our manuscript accordingly. If anything is not clear enough, please let us know.

---

> > ### Comment · Reviewer_fwDB · 2024-06-04
> > **Reviewer response**
> >
> > I thank the authors for their response. I will keep my verdict.

---

> > > ### Author Response · Authors · 2024-06-06
> > >
> > > Thank you sincerely for your time and effort on our manuscript!

---

### Official Review · Reviewer_5qEm · 2024-05-08

**Rating:** 7
**Confidence:** 3
**Ethics Flag:** 1

**Summary:**

This is a solid dataset paper, which introduces a new domain-specific text-based dataset for instruction tuning. Overall, this is a high-quality, somewhat niche, but thorough paper that shows how dataset papers ought to be done. It is very clearly written (except for a few typos here and there), provides the necessary context for a subject matter expert to understand the data generation procedure, and goes into meticulous detail of generation protocol, dataset statistics, comparison to other datasets, and experimental evaluation details. Given the importance that text-based benchmarks, even those in niche domains, have recently gained for general-purpose evaluation of LLMs, the impact of this paper is likely to be high in the community.

**Questions To Authors:**

* Q: it's unclear whether the tagging strategy used (e.g. for <SMILES>) is textual or whether it relies on a dedicated embedding for the tag (e.g. alla tagLLM https://arxiv.org/abs/2402.05140). Please clarify.
* S: label all tables, even those in the appendix, with arrows to indicate lower/higher is better
* S: there's a few typos throughout the paper, including "spitted into training" (Sec 3.2)

**Reasons To Accept:**

* Very likely will have moderate-to-high impact in the LM community, even if only "an benchmark dataset" in an LM eval suite
* Thorough description of data collection/curation pipeline
* Careful control of data leaking for reciprocal tasks (crucial for LLM benchmarks!)
* Modest but thorough baseline evaluation

**Reasons To Reject:**

* Claims dataset size as a strength, but it is only 3x compared to most similar dataset (Mol-Instructions)
* Dataset is somewhat niche / narrow

---

> ### Author Rebuttal · Authors · 2024-05-31
>
> Thank you for recognizing our work’s potential impact as “modest-to-high” and characterizing our efforts as “thorough” and “careful”. We address your concerns and questions below.
>
> ---
> **“only 3x compared to most similar dataset (Mol-Instructions)”**
>
> We clarify our efforts that lead to the size of our dataset:
> - We included all available data that we deem reliable and necessary. Low-quality sources and redundant samples are not included.
> - We removed many duplicate and low-quality samples, which decreased the quantity but expectedly enhanced the overall quality.
>
> Therefore, we believe our dataset size is a strength, given that these 3X data samples are of higher quality overall.
>
> ---
> **“dataset is somewhat niche/narrow”**
>
> Our dataset contains 14 chemistry tasks focused on small molecules. While this may seem niche, we would like to stress on the profound impact of small molecules and why we intentionally design our dataset in this way:
> - We chose to focus on small molecules due to their crucial role in critical fields like drug discovery and material science. This specialization ensures our dataset is highly relevant and allows us to provide deeper, more applicable insights into those fields.
> - The 14 tasks were selected for their real-world significance, particularly in drug discovery, and the availability of high-quality data. Each task addresses key challenges in the field, ensuring the dataset's practicality and value. Tasks lacking sufficient high-quality data, like yield prediction, are excluded to maintain quality.
>
> ---
> **Q1: Clarification about the special tags.**
>
> The special tags are implemented textually, inserted directly into query/response sentences. Their embeddings are derived using the default tokenizer of the base models. We will add the clarification in our manuscript.
>
> ---
> **Q2: Using arrows to indicate lower/higher.**
>
> Thanks. We will add arrows as you suggested.
>
> ---
> **Q3: Typos.**
>
> Thanks. We will carefully fix the typographical errors.
>
> ---
> Thank you again for your time and effort. If anything is not clear, please feel free to let us know.

---

> > ### Comment · Reviewer_5qEm · 2024-06-04
> > **Thank you for the responses**
> >
> > Thank you for the responses and clarifications. My assessment of the strengths/weaknesses of the paper remains the same. I still think the dataset's scope is narrow (which does not mean it's not significant/relevant - my original review does not put that in doubt).

---

> > > ### Author Response · Authors · 2024-06-06
> > >
> > > Understood. Thank you very much!

---

### Official Review · Reviewer_Qxrz · 2024-05-09

**Rating:** 5
**Confidence:** 5
**Ethics Flag:** 1

**Summary:**

This paper proposes SMolInstruct, a large-scale instruction-tuning dataset designed with over 3 million samples and 14 chemistry tasks. The authors use this dataset to finetune several open-source LLMs like Galactica, LLaMA, and Mistral, creating the LlaSMol series of models tailored for chemistry applications. Extensive experiments show that the LlaSMol models significantly outperform existing LLMs like GPT-4 and Claude on the chemistry tasks, though they still trail behind task-specific models. The authors conduct analyses on different modeling choices like canonicalization of SMILES, use of SELFIES vs SMILES representations, and comparison to a previous chemistry instruction dataset.

**Questions To Authors:**

I don't have additional questions for the authors.

**Reasons To Accept:**

1. SMolInstruct is large enough and diverse enough for instruction tuning of LLMs, with 3 million carefully filtered samples over 14 downstream tasks.
2. Four backbone LLMs of different architectures are evaluated in the experiments of this paper, and the performance gains are significant.
3. Careful dataset splitting avoids potential data leakage across related tasks.

**Reasons To Reject:**

1. All the data collection, validation, and processing techniques have been used by previous studies. The novelty of this paper is limited.
2. The evaluation metrics like exact match do not fully capture the models' capabilities. For example, this results in too many zeros in Table 1. Some better metrics may help reveal the models' performances.
3. The paper does not explore the models' generalization abilities beyond the specific downstream tasks in the dataset.
4. While outperforming existing LLMs, the LlaSMol models fail to surpass the non-LLM models which have far fewer parameters.

---

> ### Author Rebuttal · Authors · 2024-05-31
>
> Thank you for characterizing our work as “large/diverse dataset”, “significant performance gains”, and “careful data splitting”. We address your concerns below.
>
> ---
> **“novelty is limited”**
>
> “All the techniques have been used by previous studies” is actually not true. Our novelties:
> - For the first time, PubChem and MoleculeNet are used for instruction tuning, which is not trivial.
> - We introduce novel methods tailored for chemistry. To name a few, we use specialized methods to eliminate duplicates and data leakage, and introduce SMILES canonicalization and special tags.
> - We are the first to comprehensively train and evaluate multiple LLMs on chemistry. Our models outperform existing LLMs, and experiments on task synergies, varying trainable parameters, and more provide significant insights.
>
> ---
> **“evaluation metrics do not fully capture models’ capabilities”**
>
> We use EM as it directly measures the correctness, which is crucial for chemistry. As described in Appendix D.4, EM evaluates on a deeper semantic level instead of a simple string comparison, and thus can effectively capture model performance. Also, EM is widely used in prior research. The zeros in Table 1 actually reflect the models’ incompetence when not turned on our dataset, not a flaw in EM.
>
> For a comprehensive evaluation, we have included many other metrics such as validity and FTS (Appendix D.4 and E). We welcome any other metrics you might recommend.
>
> ---
> **“does not explore generalization abilities”**
>
> We included all tasks we can find high-quality data for, and building models that perform well on all of them is a significant contribution. How to meaningfully test generalization abilities is nontrivial and needs careful design, and we leave it for future work.
>
> ---
> **“fail to surpass the non-LLM models with far fewer parameters”**
>
> There seems to be a misunderstanding. Due to resource constraints, we only tune 42M (0.58%) of LlaSMol's parameters, significantly fewer than the non-LLM models (e.g., MT 150M, Uni-Mol 181M). Despite this, LlaSMol approaches their performance levels, which is encouraging. Also, our exp on varying trainable parameters (Appendix F.2) highlights our potential to surpass non-LLM models with further tuning.
>
> Beyond performance, LlaSMol offers unique advantages over non-LLM models, such as handling multiple tasks and following natural language instructions to complete tasks.
>
> ---
> Thank you for your time and effort. If anything is not clear, please let us know.

---

> > ### Author Response · Authors · 2024-06-04
> >
> > Thank you sincerely for your time on our manuscript. We hope we addressed your concerns clearly. If there is anything still unclear to you, please do not hesitate to let us know :)

---

> > ### Comment · Reviewer_Qxrz · 2024-06-06
> > **Feedback**
> >
> > ```
> > For the first time, PubChem and MoleculeNet are used for instruction tuning, which is not trivial.
> > ```
> >
> > **Response:** PubChem is used for instruction tuning in 3D-MoLM [1]; MoleculeNet is used in Mol-Instructions [2].
> >
> > ```
> > Some relevant works are not appropriately acknowledged in the related work:
> > ```
> >
> > * *molca: molecular graph-language modeling with cross-modal projector and uni-modal adapter. In EMNLP 2023.* **It seems LlaSMol share many similar tasks with this earlier work, including 1) presents property prediction results on the MoleculeNet datasets (e.g., BBBP, Clintox, and SIDER); 2) presents name conversion results;  and 3) present molecule caption results**
> > * *Towards 3D Molecule-Text Interpretation in Language Models. In ICLR 2024.* This earlier work also employs Uni-Mol for molecule property prediction, and shares some other tasks.
> >
> > **Why are these relevant works not discussed or compared?**
> >
> >
> >
> > **Reference:**
> >
> > [1] Towards 3D Molecule-Text Interpretation in Language Models
> >
> > [2] Mol-Instructions: A Large-Scale Biomolecular Instruction Dataset for Large Language Models

---

> > ### Author Response · Authors · 2024-06-07
> >
> > Thank you for your insightful concerns.
> >
> > ---
> > **“PubChem and MoleculeNet are not used for instruction tuning for the first time”**
> >
> > Sorry for the inaccurate expression. We meant to say that they are used for the first time for many of our carefully selected tasks (and why we selected these tasks was described in Section 3.1), and we made non-trivial efforts in preparing the data for instruction tuning on such tasks. Specifically:
> >
> > - Although PubChem has been used for instruction tuning in 3D-MoLM [1], it is for the molecule-text interpretation task, instead of our name conversion tasks. To create data for name conversion, we collected and sampled compound data points from PubChem, extracted their multiple representations, and removed those data points with invalid SMILES or incomplete domains (i.e., data points with no SMILES/IUPAC/Molecular formula specified).
> > - MoleculeNet is used for instruction tuning on properties like solubility and side effects for the first time, while Mol-Instructions only uses the QM9 dataset of MoleculeNet for energy value prediction. We intentionally and carefully chose these included properties, because domain experts suggested that they are very important and related to real-world applications like drug discovery. Besides rigorously examining data quality and fixing some issues, we put a lot of effort in creating instructions for the side effect dataset (PP-SIDER): we first removed some side effect labels whose names are not clear and specific enough for instruction tuning (e.g., Investigations), and manually created the instruction questions for each of the side effects to ensure they are correct and diverse.
> >
> > In conclusion, we used PubChem and MoleculeNet for instruction tuning on many carefully selected tasks for the first time, and applied non-trivial and tailored methods to process the data. Therefore, the set of instruction tuning data created by us is novel and should contribute to the area.
> >
> > ---
> > **“Relevant works [3,1] are not appropriately acknowledged in the related work”  “these relevant works [3,1] are not discussed or compared”**
> >
> > Thanks for mentioning these relevant works. We will definitely discuss them in our updated version. We primarily focus on evaluating LLMs on chemistry, especially those generalist models that can handle multiple tasks by following natural language instructions. There are indeed many existing works for each of the tasks in our proposed benchmark, and due to paper length and resource limit, we are not able to discuss or compare with many of them, but only choose one of the most representative SoTA models for each task and report its results. That said, we agree with the reviewer, and will add more discussions on [1, 3] in our updated manuscript and compare with them on applicable tasks. We would like to gently note that [1] seems to be released at the end of Jan 2024, around two months before the COLM deadline, which is usually regarded as concurrent work.
> >
> > ---
> > We sincerely appreciate your invaluable feedback, and will improve our manuscript accordingly. Thank you!
> >
> > ---
> > **Reference**
> >
> > [1] Li, S., Liu, Z., Luo, Y., Wang, X., He, X., Kawaguchi, K., ... & Tian, Q. (2024). Towards 3D Molecule-Text Interpretation in Language Models. ICLR 2024.
> >
> > [2] Fang, Y., Liang, X., Zhang, N., Liu, K., Huang, R., Chen, Z., ... & Chen, H. (2023). Mol-instructions: A large-scale biomolecular instruction dataset for large language models. In ICLR 2024.
> >
> > [3] Liu, Z., Li, S., Luo, Y., Fei, H., Cao, Y., Kawaguchi, K., ... & Chua, T. S. (2023). Molca: Molecular graph-language modeling with cross-modal projector and uni-modal adapter. In EMNLP 2023.

---

### Decision · Program_Chairs · 2024-07-10

**Decision:**

Accept

**Comment:**

In this paper, the authors propose a large-scale chemistry dataset for instruction tuning, encompassing 14 selected chemistry tasks and over three million samples. Using this dataset, the authors fine-tune a set of open-source LLMs and identify Mistral as the best base model for chemistry tasks, which outperforms the most advanced models like GPT-4 and Claude 3 Opus by a significant margin. Although one reviewer expressed concern about some inaccurate claims made by the authors and the lack of discussion on related works, the authors have committed to clarifying these claims and including such a discussion in their revisions. Given the overall strong support from the majority of reviewers, I recommend acceptance.